# Life-long impairment of glucose homeostasis upon prenatal exposure to psychostimulants

Solomiia Korchynska[1,†], Maria Krassnitzer[1,†], Katarzyna Malenczyk[1,†], Rashmi B Prasad[2], Evgenii O Tretiakov[1], Sabah Rehman[1], Valentina Cinquina[1], Victoria Gernedl[3], Matthias Farlik[3], Julian Petersen[1], Sophia Hannes[1], Julia Schachenhofer[1], Sonali N Reisinger[4], Alice Zambon[4], Olof Asplund[2], Isabella Artner[5,6], Erik Keimpema[1], Gert Lubec[7], Jan Mulder[8], Christoph Bock[3], Daniela D Pollak[4], Roman A Romanov[1], Christian Pifl[1], Leif Groop[2,9], Tomas GM Hökfelt[10] & Tibor Harkany[1,10,*] [ID]

## Abstract

Maternal drug abuse during pregnancy is a rapidly escalating societal problem. Psychostimulants, including amphetamine, cocaine, and methamphetamine, are amongst the illicit drugs most commonly consumed by pregnant women. Neuropharmacology concepts posit that psychostimulants affect monoamine signaling in the nervous system by their affinities to neurotransmitter reuptake and vesicular transporters to heighten neurotransmitter availability extracellularly. Exacerbated dopamine signaling is particularly considered as a key determinant of psychostimulant action. Much less is known about possible adverse effects of these drugs on peripheral organs, and if *in utero* exposure induces life-long pathologies. Here, we addressed this question by combining human RNA-seq data with cellular and mouse models of neuroendocrine development. We show that episodic maternal exposure to psychostimulants during pregnancy coincident with the intrauterine specification of pancreatic β cells permanently impairs their ability of insulin production, leading to glucose intolerance in adult female but not male offspring. We link psychostimulant action specifically to serotonin signaling and implicate the sex-specific epigenetic reprogramming of serotonin-related gene regulatory networks upstream from the transcription factor *Pet1/Fev* as determinants of reduced insulin production.

**Keywords** diabetes; gender; serotonin; susceptibility; β cell
**Subject Categories** Metabolism; Neuroscience

**The EMBO Journal (2020) 39: e100882**

## Introduction

The misuse of psychostimulants is a major socioeconomic burden, affecting up to 50 million people worldwide (Oliere *et al*, 2013). Population analyses show that 5.4% of pregnant females aged between 15 and 44 years abuse illicit drugs, highlighting fetal exposures as a significant public concern (Substance Abuse and Mental Health Services Administration, 2014). Amphetamine, cocaine, and methamphetamine are the most commonly consumed psychostimulants with their peripheral effects mimicking the activation of the sympathetic nervous system: increased heart rate and blood pressure, decreased digestion, and mobilization of glucose reserves (Favrod-Coune & Broers, 2010). Their mechanism of action centers on molecular interference with monoamine reuptake transporters (i.e., reversal of substrate flow) (Howell & Negus, 2014; Ross *et al*, 2015) and/or inhibition of monoamine oxidases. Consequently, extracellular concentrations of dopamine (DA), norepinephrine (NE), and serotonin (5-HT) surge, which prolongs their action on respective receptor systems (Kitanaka *et al*, 2006; Ross *et al*, 2015; Liu *et al*, 2016).

Amphetamine, cocaine, and methamphetamine can cross the placenta to directly affect the developing fetus (Thompson *et al*,

1  Department of Molecular Neurosciences, Center for Brain Research, Medical University of Vienna, Vienna, Austria
2  Department of Clinical Sciences, Diabetes and Endocrinology CRC, Skåne University Hospital Malmö, Malmö, Sweden
3  CeMM Research Center for Molecular Medicine of the Austrian Academy of Sciences, Vienna, Austria
4  Department of Neurophysiology and Neuropharmacology, Center for Physiology and Pharmacology, Medical University of Vienna, Vienna, Austria
5  Stem Cell Center, Lund University, Lund, Sweden
6  Endocrine Cell Differentiation and Function, Lund University Diabetes Center, Lund University, Malmö, Sweden
7  Paracelsus Medical University, Salzburg, Austria
8  Science for Life Laboratory, Karolinska Institutet, Solna, Sweden
9  Institute for Molecular Medicine Finland (FIMM), Helsinki University, Helsinki, Finland
10 Department of Neuroscience, Karolinska Institutet, Solna, Sweden
   *Corresponding author. Tel: +43 1 40160 34050; Fax: +43 1 40160 934053; E-mail: tibor.harkany@meduniwien.ac.at
   †These authors contributed equally to this work

2009). Given their potent action on brain function and development (Ross *et al*, 2015), the bulk of available literature focuses on psychostimulant effects on the behavior and cognition of affected offspring (Middaugh, 1989; Diaz *et al*, 2014; Volkow & Morales, 2015; Martin *et al*, 2016). Even though a link between prenatal psychostimulant exposure and an increased risk for the offspring to develop type 2 diabetes has been noted and associated with maternal malnutrition (Vaiserman, 2015), data on the psychostimulant-induced priming of glucose metabolism to render affected offspring glucose intolerant are lacking. This is unexpected because endocrine cells of the pancreas are programmed during embryonic and early postnatal periods and, consequently, only commence hormone secretion postnatally (Murtaugh, 2007). Thus, disrupted cellular turnover and inter- or intracellular signaling for terminal differentiation during intrauterine development carries the risk of lifelong impairments of bodily energy homeostasis.

DA, NE, and 5-HT signaling regulate secretory functions of the endocrine pancreas. DA and its precursor L-dopa inhibit glucose-stimulated insulin secretion (Ustione & Piston, 2012). NE released from innervating sympathetic nerves shapes islet architecture and inhibits insulin but promotes glucagon release (Sorenson *et al*, 1979; Borden *et al*, 2013). 5-HT is proposed to function as a paracrine signal stimulating β cell proliferation and insulin release, while inhibiting glucagon secretion (Paulmann *et al*, 2009; Kim *et al*, 2010; Almaca *et al*, 2016). 5-HT action on β cells converges on *Pet1* (alternative name: *Fev*), an E-twenty-six (ETS) transcription factor that, in analogy to being a "master transcriptional regulator" of 5-HT programs in neurons (Bonnin & Levitt, 2011), induces insulin (*Ins1/2*) expression in a mechanism downstream from, e.g., 5-HT1A (*Htr1a*) receptors (Ohta *et al*, 2011). Therefore, we hypothesized that fetal β cell programming by monoamines, particularly 5-HT, can be susceptible to maternal psychostimulant abuse with adverse effects enduring into the adulthood of affected offspring.

Here, we interrogated data from RNA-seq of the fetal human pancreas with support from single-cell RNA-seq of human α and β cells to show a molecularly complete (cell-autonomous) metabolic loop for 5-HT but neither DA nor NE signaling. Accordingly, β cells express both SLC18A2 and SLC6A4 (SERT) but not SLC6A3/SLC6A2, highlighting preferential 5-HT transport as a critical molecular substrate for psychostimulant action *in utero*. 5-HT signaling and uptake operate in both fetal and adult pancreata of mice and are impaired by psychostimulants. We provide evidence that psychostimulants acutely reshape β cell excitability in a fashion that is sensitive to competition by escitalopram, a selective SERT inhibitor. We then find that 5-HT uptake primes protein serotonylation in a β cell model *in vitro*, which is eliminated by psychostimulants, e.g., amphetamine. Subsequently, a lifelong reduction of 5-HT-driven *Pet1/Fev* expression is identified, which correlates with that of insulin and 5-HT in pancreatic β cells prenatally exposed to psychostimulants. These molecular changes are sufficient to compromise glucose homeostasis for life with female offspring in experimental models being more susceptible to developing pre-diabetic glucose intolerance by adulthood than males. However, it is neither *Pet1/Fev* not *Ins1/2* itself but their upstream 5-HT-sensitive gene regulatory networks that undergo lifelong epigenetic reprogramming in the prenatally

psychostimulant-exposed pancreas. In sum, these data uncover key molecular determinants of permanent pancreas dysfunction in offspring from mothers with a history of drug abuse during pregnancy.

## Results

### Monoamine signaling in the human fetal pancreas

Given that the widely accepted mechanism of action for psychostimulants is interference with both intracellular vesicular transport and cell-surface reuptake systems tuning monoamine levels extracellularly (Ross *et al*, 2015), we first profiled molecular constituents of 5-HT, DA, and NE signaling in fetal human pancreata (Fig 1). These data showed mRNA expression for 5-HT receptors (HTR1B, HTR2A/B/C, HTR3E), the 5-HT reuptake transporter SLC6A4 (SERT), and tryptophan hydroxylases (particularly TPH1; Fig 1A). During mid-to-late 1st trimester of pregnancy, the fetal human pancreas also expresses adrenergic receptors (ADRA2A/2B, ADRA3B), dopamine-β-hydroxylase (DBH), and more infrequently SLC6A2 (NET), mediating NE reuptake (Fig 1C). Notably, neither dopamine receptors nor SLC6A3 (DAT) are expressed even if tyrosine hydroxylase (TH) is detected at varying levels (Fig 1D). Expression of SLC18A2, encoding the vesicular monoamine transporter (VMAT2), supports

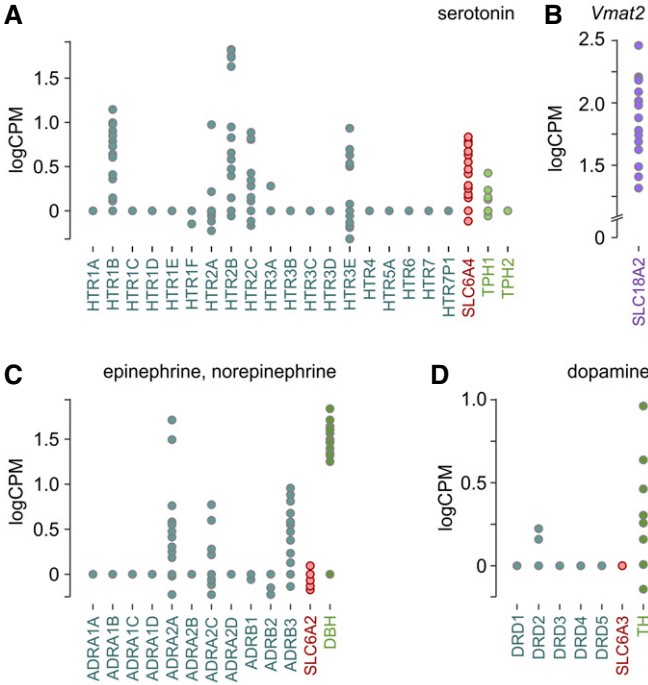

**Figure 1. Monoamine signaling in human fetal pancreas.**

A–D   RNA-seq of fetal pancreata during gestational weeks 7–14 revealed the presence (or lack) of molecular determinants for serotonin (A), epinephrine/norepinephrine (C), and dopamine (D) signaling. Expression of the vesicular monoamine transporter (SLC18A2; *Vmat2*), which is non-selective for monoamines, was plotted separately (B). Data were expressed as log counts per million mapped reads (logCPM).

efficient monoamine release (Fig 1B). These data suggest that the fetal pancreas can produce and release monoamines, in an order of 5-HT ~ NE > DA, with DA signaling being an unlikely target for psychostimulants given the lack of SLC6A3 expression. However, this analysis lacks resolution to differentiate between exocrine and endocrine components.

Therefore, we processed single-cell RNA-seq matrices of dissociated human α and β cells (Segerstolpe et al, 2016), which showed β cells to express the 5-HT receptors HTR1F and HTR2B, SLC6A4, and TPH2 (Fig EV1A). A similar pattern of expression for 5-HT signaling was also seen in human α cells (Fig EV1B). Human β cells also express a variety of adrenergic receptors (ADRA1B, ADRA2A, ADRB1, ADRB2), infrequently DBH but lack SLC6A2 (Fig EV1A1). Human α cells express ADRB1 and ADRB2 receptors abundantly, DBH infrequently with SLC6A2 expression being absent (Fig EV1B1). For DA signaling, β cells express the biosynthesis enzyme tyrosine hydroxylase (TH) but lack both SLC6A3 and receptors (Fig EV1A2). In contrast, α cells express DRD1 and DRD4 receptors but neither TH nor SLC6A3 (Fig EV1B2). Expression of SLC18A2 (VMAT2) supports efficient monoamine release from both β and α cells (Fig EV1A3 and B3). These data are congruent with our fetal tissue profiling and suggest that pancreatic β and α cells harbor the capacity of both cell-autonomous and bidirectional paracrine 5-HT signaling (Almaca et al, 2016). In addition, both cell types can sense extracellular NE and utilize unidirectional β-to-α cell communication by DA. Considering that β cells express a functional 5-HT signaling cassette, and 5-HT signaling was previously linked to β cell proliferation and insulin and glucagon secretion (Paulmann et al, 2009; Kim et al, 2010; Almaca et al, 2016), we addressed the organization of 5-HT signaling in the endocrine pancreas of developing mice. Given the presence of SLC6A4 but not SLC6A2/SLC6A3 in β cells, this was followed by interrogating psychostimulant-induced modifications to 5-HT signaling.

### 5-HT signaling in developing mouse pancreas

To gain insights into the organization of 5-HT signaling in the endocrine pancreas, we histochemically sampled mouse tissues at embryonic day (E)14.5 (which corresponds to the start of the secondary transition that generates endocrine lineages (Mastracci & Sussel, 2012)), birth (P0), and adulthood. 5-HT was not detected and other markers were lowly expressed at E14.5. Therefore, we have focused on neonatal (P0) and 6-week time points. All molecular determinants of 5-HT signaling showed a preference toward expression in insulin-positive($^+$) β cells (Fig 2A–J1), including TPH2 (Fig 2A and B1), 5-HT (Fig 2C and D2), SERT (Fig 2E and F1), 5-HT receptor 1F (5-HTR$_{1F}$; Fig 2G and H1), and 5-HTR$_{1A}$ (Fig 2I and J1). The expression of TPH2 and SERT, which were stably seen in β cells, was also confirmed by qPCR (Appendix Fig S1A). Even though we tested a number of anti-VMAT2 antibodies, none of these produced reliable immunohistochemical signals, which corroborate the reported lack of VMAT2 expression in laboratory rodents (Schafer et al, 2013). These data are compatible with the notion that 5-HT signaling is a critical factor for pancreas development (Ohta et al, 2011) and show that 5-HTR$_{1F}$ and 5-HTR$_{1A}$ can support 5-HT signaling in mice (Almaca et al, 2016).

Next, we used INS-1E cells, an in vitro model of insulin secretion (Asfari et al, 1992; Malenczyk et al, 2013), to test their

responses to 5-HT by Ca$^{2+}$ imaging particularly since Ca$^{2+}$ peaks are considered reliable indicators of glucose-induced β cell excitability (Bergsten et al, 1994; Fridlyand et al, 2010). In 11 mM glucose, INS-1E cells were spontaneously active (i.e., they release insulin through regulated exocytosis; Malenczyk et al, 2017; Fig 2K). 5-HT in the 100 nM-to-1 μM range induced a dose-dependent reduction of the frequency of Ca$^{2+}$ oscillations (Fig 2K1). At the same time, the amplitude of Ca$^{2+}$ oscillations significantly increased (Fig 2K1). These data show that INS-1E cells respond to 5-HT bath-applied acutely and suggest that subthreshold and fast Ca$^{2+}$ events appear as high-amplitude oscillations by Fura2-AM-based Ca$^{2+}$ imaging.

We then asked whether INS-1E cells could also respond to DA and NE. By using $^3$H-labeled monoamines, we find considerable 5-HT but neither DA nor NE uptake (Fig 2L). [$^3$H]5-HT uptake was significantly inhibited by escitalopram, confirming SERT involvement (Fig 2L). Notably, the level of SERT activity in INS-1E cells was a magnitude lower than seen in serotonergic neurons in brain (Ohta et al, 2011). Subsequently, we showed that 5-HT uptake lacks a reserpine-sensitive component, confirming that VMAT2 is unlikely to play a role in this uptake mechanism (Fig 2M). These data thus highlight the preferential use of 5-HT by β cell-like cells and corroborate our and others' (Schafer et al, 2013) data on the absence of VMAT2 in rodent β cells. Thus, we suggest a role for 5-HT upon SERT-mediated uptake as an intracellular precursor rather than a transmitter directly co-partitioned with insulin into release granules (Fig 2N).

### Psychostimulants induce inward currents upon SERT engagement and inhibit Ca$^{2+}$ signaling

The presence of a molecularly intact 5-HT signaling loop in β cells together with their 5-HT sensitivity suggests that β cells could be equally sensitive to illicit psychostimulants whose mechanism of action centers on SERT inactivation by phosphorylation and sequestration (Ramamoorthy & Blakely, 1999) and even reversal of its electrogenic transport (Shi et al, 2008; Robertson et al, 2009; Sitte & Freissmuth, 2015). Psychostimulants were previously shown to induce inward currents in neurons and cultured cell lines (Hilber et al, 2005), which could be explained by increased intracellular Na$^+$ co-transported by SERT (Robertson et al, 2009). To confirm this hypothesis, we recorded psychoactive drug effects in HEK293 cells stably expressing the human SERT (HEK293-SERT) clamped to a holding potential of −70 mV (Fig 3A). Amphetamine (25 μM), cocaine (10 μM), and methamphetamine (5 μM) induced significant inward currents, which appeared with considerable time-lag for amphetamine and cocaine and were escitalopram (10 μM)-sensitive (Fig 3A). Escitalopram alone did not affect the membrane potential of HEK293-SERT cells (Fig 3A1).

Subsequently, we exposed INS-1E cells to increasing concentrations of the psychostimulants applied acutely by superfusion and recorded their intracellular Ca$^{2+}$ responses. INS-1E cells did not produce spontaneous Ca$^{2+}$ oscillations in the presence of low glucose (3 mM; Fig EV2A–D). None of the drugs (amphetamine, cocaine or methamphetamine) used exerted any effect either (Fig EV2A–D). In contrast, superfusion of 11 mM glucose induced Ca$^{2+}$ oscillations in INS-1E cells with both their frequency and

amplitude reduced by amphetamine at doses > 10 μM (Fig 3B; see also Fig EV2A1). Cocaine induced a biphasic effect: up to 10 μM concentration, it reduced the frequency, but not the amplitude of $Ca^{2+}$ waves. At a dose of 100 μM, however, it augmented both the amplitude and the frequency of $Ca^{2+}$ oscillations ($P < 0.01$; Figs 3B and EV2B1). Methamphetamine effects were reminiscent to those of amphetamine (at a magnitude lower concentration range; see

Fig EV2C1), reducing the frequency of $Ca^{2+}$ oscillations in a dose-dependent fashion.

Psychostimulant action could be indirectly mediated by either the allosteric modulation of 5-HT receptors (Borycz et al, 2008; Zhou & Cunningham, 2018) or the buildup of extracellular 5-HT and its spillover upon SERT occlusion. Here, we tested whether escalating doses of escitalopram evoke responses similar to those by the drugs used.

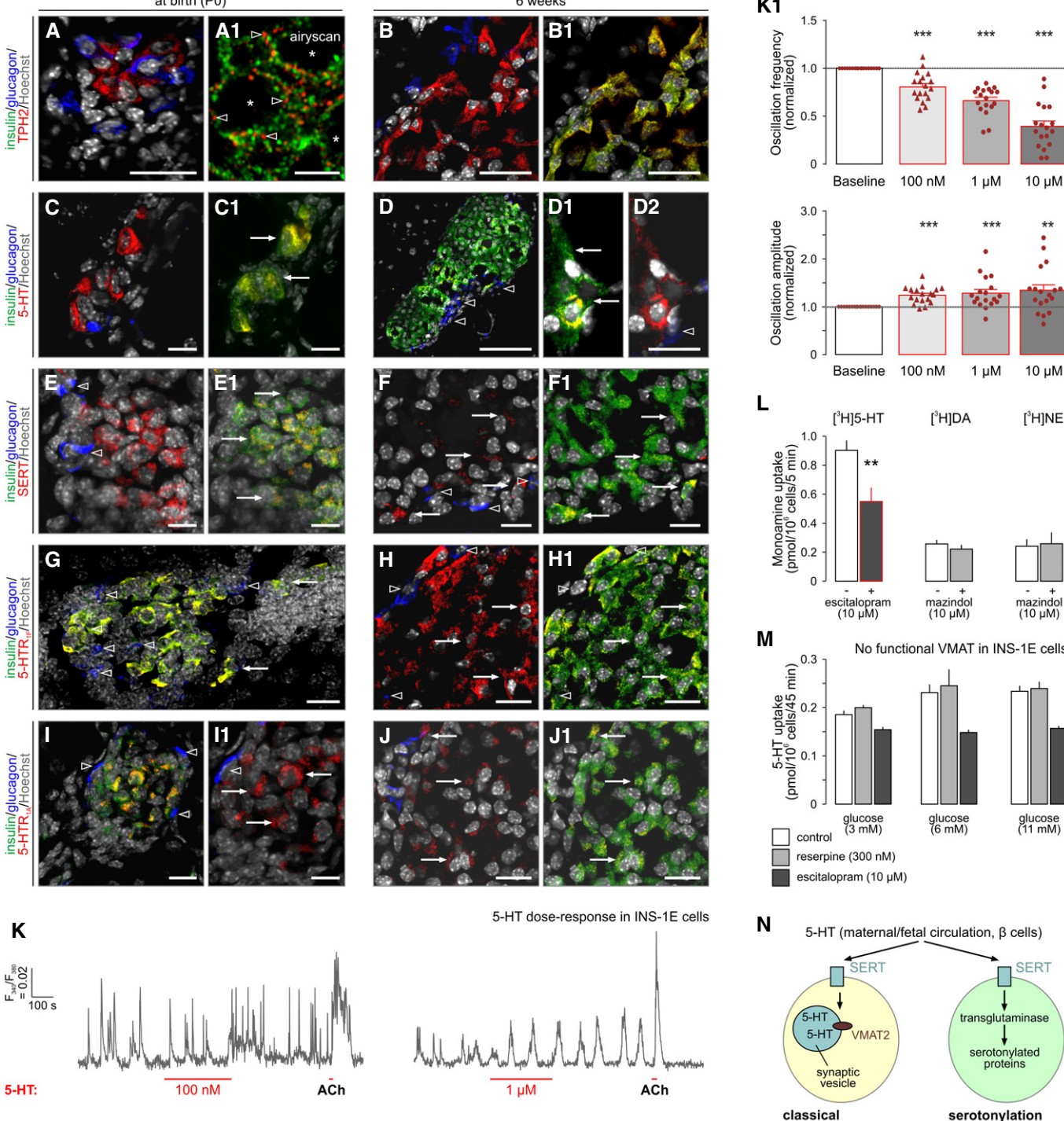

Figure 2.

**Figure 2.  Serotonin signaling during development of the endocrine pancreas in mouse.**

A–J  All markers are shown at birth (P0) and at 6 weeks of age: (A-B1) tryptophan hydroxylase 2 (TPH2), (C-D2) serotonin (5-HT), (E-F1) serotonin reuptake transporter (SERT), (G-H1) 5-HT receptor 1F subtype (5-HT1F), and (I-J1) 5-HT1A. Note that all histochemical marks localized to β cells with only sporadic expression, if any, in α cells. 5-HT was mostly absent in E14.5 pancreata therefore earlier developmental stages are not shown. "*Airyscan*" (A1) identifies the lack of false-positive co-localization between TPH2 and insulin at superresolution. Open arrowheads indicate the lack of co-localization between select markers. Solid arrows pinpoint cellular sites of co-existence for pairs of histochemical marks. Asterisks label the positions of nuclei (A1). Representative images are shown from $n \geq 3$ pancreata/age group. Hoechst 33342 was used as a nuclear counterstain (pseudo-colored in gray). *Scale bars* = 80 μm (D), 20 μm (A–B1,F–J1), 10 μm (C–E1), and 1 μm (A1).

K  Representative $Ca^{2+}$ signals upon exposure of INS-1E cells pre-loaded with Fura2-AM to 5-HT at the concentrations indicated. Drugs were superfused acutely as shown. Acetylcholine (ACh, 5 μM) served as positive control. (K1) Quantitative analysis of $Ca^{2+}$ responses in INS-1E cells to incrementing concentrations of 5-HT. Data on the frequency (*upper*) and amplitude (*lower*) of $Ca^{2+}$ transients are shown (normalized to baseline).

L  Tritiated monoamine uptake in INS-1E cells and its sensitivity to selective SERT and DAT/NET inhibitors.

M  5-HT uptake in INS-1E cells in the presence of increasing extracellular glucose concentrations. Reserpine was used to inhibit VMAT2.

N  Schema outlining intracellular signaling and metabolic cascades in fetal β cells for the use of 5-HT taken up through SERT engagement. "Classical" refers to a neuron-like/adult-like scenario with 5-HT directly loaded into release vesicles by VMAT2 (or a hypothetical alternative transport mechanism).

Data information: Data in (K1–M) were expressed as means ± SEM. Red symbols in K1 mark individual observations, that is, biological replicates. ***$P < 0.001$, **$P < 0.01$ (pair-wise comparisons after one-way ANOVA).

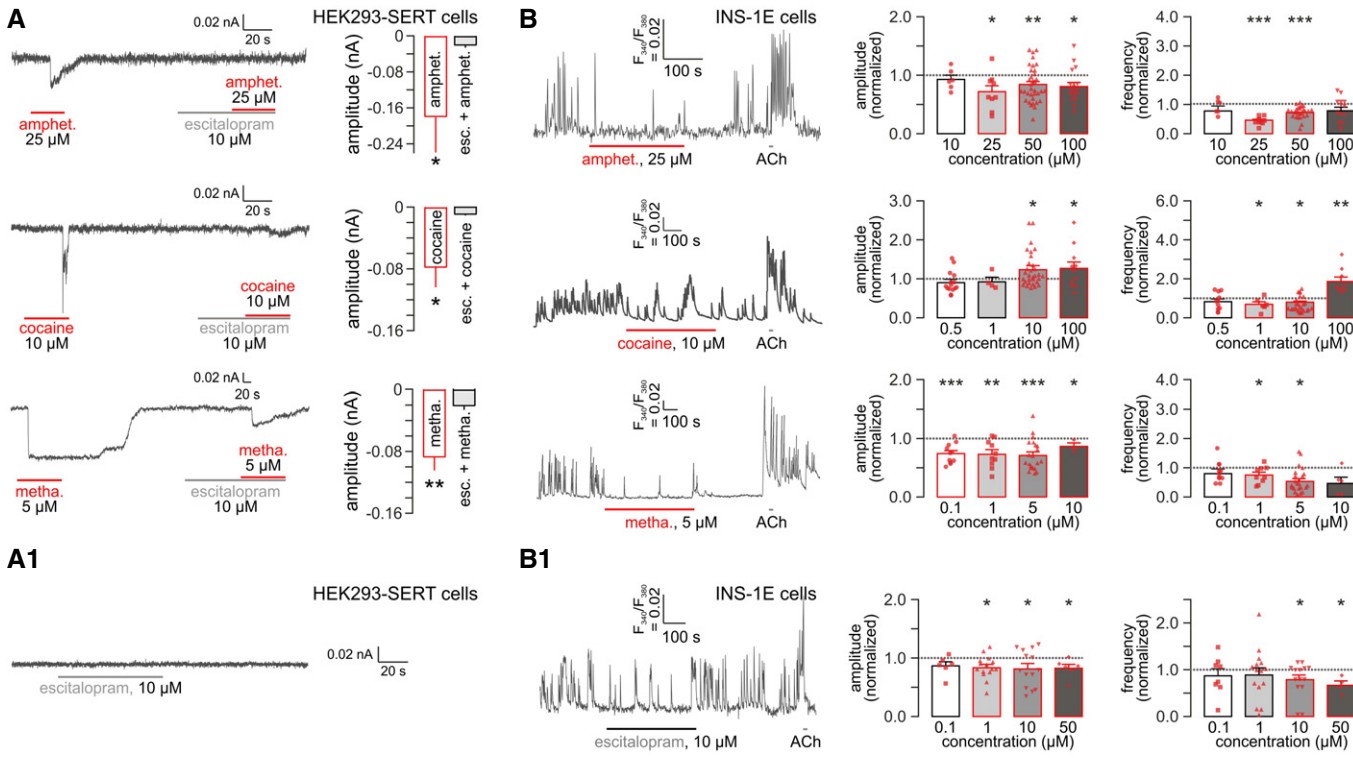

**Figure 3.  Psychostimulants induce electrogenic currents and modulate $Ca^{2+}$ signaling in INS-1E cells.**

A  Psychostimulants (A) but not escitalopram (A1) induced inward currents in HEK293 cells transfected with the human serotonin transporter (HEK293-SERT). Note that escitalopram co-application occluded psychostimulant effects. For each drug, a representative recording trace is shown on the *left*, while quantitative data (means ± SEM; $n > 6$/group) are on the *right*. Note that cocaine effects were short-lived downward deflections, whereas methamphetamine commonly induced prolonged inward currents that lasted even during washout. HEK293 cells were clamped at −70 mV throughout.

B  Amphetamine and methamphetamine reduced both the amplitude and frequency of $Ca^{2+}$ oscillations in INS-1E cells loaded with Fura-2AM. In contrast, cocaine at high concentrations increased both parameters measured. Data were normalized to baseline and expressed as means ± SEM. Symbols in red denote biological replicates. (B1) Escitalopram alone also reduced the amplitude and slowed the frequency of $Ca^{2+}$ oscillations in INS-1E ells. Experiments were performed in the presence of 11 mM glucose. Acetylcholine (ACh; 5 μM) was used as positive control. Figure EV3 is referred to showing representative $Ca^{2+}$ recordings for drug concentrations other than the ones on the *left* of (B, B1).

Data information: ***$P < 0.001$, **$P < 0.01$, *$P < 0.05$ (pair-wise comparisons after one-way ANOVA).

Indeed, acute escitalopram superfusion of INS-1E cells at 10 μM or higher concentrations also reduced both the amplitude and frequency of $Ca^{2+}$ oscillations ($P < 0.05$; Figs 3B1 and EV2D1).

We then superfused escitalopram (1 μM) alone or in combination with the psychostimulants. Because escitalopram binds to the orthosteric site of SERT, only partially, if at all, overlapping

(Kristensen *et al*, 2011) with amphetamine, cocaine, and methamphetamine bindings sites, we expected that subthreshold escitalopram doses could sensitize INS-1E cells to psychostimulants. Indeed, the psychostimulant-induced slowing of $Ca^{2+}$ oscillations was significantly augmented by escitalopram (Fig EV3). A rebound effect was seen for the amplitude of $Ca^{2+}$ transients in the presence of 50 µM amphetamine though (Fig EV3A). For cocaine (Fig EV3B) and methamphetamine (Fig EV3C), however, the amplitudes of $Ca^{2+}$ oscillations were reduced in a dose-dependent manner. These data cumulatively suggest that psychostimulants engage SERT, and can cooperate with escitalopram to modulate intracellular $Ca^{2+}$ signaling in INS-1E cells *in vitro*.

### Amphetamine reduces protein serotonylation *in vitro*

Blockade, and even reversal, of cell-surface transporters mediating monoamine reuptake (NET, DAT, SERT) is one of the mechanisms by which psychostimulants enhance extracellular monoamine levels (Pifl *et al*, 1995; Steinkellner *et al*, 2014; Sitte & Freissmuth, 2015). Therefore, whether SERT is operational in fetal pancreas is of mechanistic significance. We addressed this question by preparing pancreas explants on E13.5 and culturing those for 6 days. To closely match the *in vivo* experimental conditions (Appendix Fig S2), explants were treated with 5-HT (500 nM) daily 1–3 days later, equivalent to the period of E14.5-16.5. Twenty-four hours after the last treatment, pancreata were transferred to fresh medium and cultured for another 2 days. 5-HT accumulated in pancreas explants as shown by both immunofluorescence cytochemistry (Fig 4A) and HPLC (in INS-1E cell homogenates 45 min after extracellular 5-HT (5 µM) loading; Fig 4B) (Pifl *et al*, 1995). These data demonstrate that 5-HT uptake mechanisms are likely operational during fetal development with intracellular 5-HT available over considerable periods. More broadly, these results suggest that the fetal pancreas can be sensitive to blood-borne 5-HT produced by the placenta (from maternal tryptophan precursors) or other maternal sources (Cote *et al*, 2007; Bonnin *et al*, 2011).

We then asked whether intracellular 5-HT retained over prolonged periods is of functional significance. In INS-1E cells, 5-HT exposure for 4 h significantly increased the level of proteins that had incorporated a 5-HT moiety (Fig 4C), indicative of protein serotonylation. Notably, and regardless of the molecular identity of the target proteins (*note that Rabs were assigned as preferred molecular targets in β cell earlier (Paulmann et al, 2009)*), amphetamine occluded protein serotonylation when co-applied with 5-HT (Fig 4C). These data reveal that long-lived 5-HT signals in fetal pancreata might indicate the accumulation of transamidated proteins covalently incorporating 5-HT during β cell development. Moreover, they corroborate recent data in $Tph1^{-/-}$ mice, which present disrupted protein serotonylation in the pancreas as part of their diabetic pathology (Paulmann *et al*, 2009).

### Intrauterine psychostimulant exposure reduces 5-HT and insulin content in fetal pancreas

We tested the effect of psychostimulants on endocrine pancreas development *in vivo* by injecting (*i.p.*) pregnant dams with amphetamine (10 mg/kg body weight), cocaine (20 mg/kg), methamphetamine (10 mg/kg), or saline (as control) once daily between E14.5 and 16.5 (Fig 4D), corresponding to drug doses commonly used to disrupt brain development (Lee & O'Dowd, 1999; Khoradmehr *et al*, 2015). None of the pregnancies were aborted, and litter sizes were not affected by drug treatment [$7 \pm 2$ (methamphetamine), $7 \pm 1$ (amphetamine), $7 \pm 1$ (cocaine) versus $6 \pm 1$ (saline)]. At P0, each litter was reduced to 6 pups, which were used to analyze prenatal drug effects on postnatal glucose metabolism.

Amphetamine, cocaine, and methamphetamine exert cytotoxic effects in the brain (Zhu *et al*, 2006; Costa *et al*, 2013). Therefore, we first determined the number of β and α cells in P0 pancreata. None of the drugs reduced either β or α cell numbers (Fig 4D1 and D2), suggesting that the doses of psychostimulants used did not *per se* disrupt pancreas development. Nevertheless, the psychostimulants used significantly reduced intracellular 5-HT content in β cells, measured immunohistochemically [$P < 0.01$ (amphetamine), $P < 0.05$ (cocaine), $P < 0.01$ (methamphetamine) versus (saline); Fig 4D1 and E]. Insulin levels were similarly reduced [$P < 0.05$ (amphetamine), $P < 0.01$ (cocaine), $P < 0.001$ (methamphetamine), versus (saline); Fig 4F]. Since 5-HT is stored in the same granules as insulin in mature β cells (Lundquist *et al*, 1971) and protein serotonylation might be a precursor to insulin production (*see above*), we plotted 5-HT versus insulin immunoreactivity per β cell: all treatments returned significant positive correlation (Fig 4G) suggesting synchronous regulation of 5-HT and insulin levels. None of the psychostimulants affected glucagon content in the neonatal pancreas (Appendix Fig S1B), pinpointing β cells as a locus for adverse developmental drug effects.

The above observations together with our data from pancreatic explants would suggest that reduced insulin levels in β cells indicate their delayed development (or permanent inability to produce insulin; for *Ins1* mRNA changes, we refer to Fig EV4B). The finding that the number of pancreatic and duodenal homeobox 1 $(Pdx1)^+$ cells in P0 pancreata remained unchanged suggests that they retained the capacity to produce β cells (Fig EV4A). Yet, reduced levels of neurogenin 3-dependent expression of paired neuronal differentiation 1 (*NeuroD1*) and NK6 homeobox 1 (*Nkx6.1*), late markers of β cell specification (Malenczyk *et al*, 2018) (Fig EV4B), are suggestive of delayed pancreas maturation upon preceding amphetamine (and more broadly psychostimulant) action.

Most illicit drugs exhibit gender bias: developmental toxicology studies show male sensitivity to cannabis (Jutras-Aswad *et al*, 2009). In contrast, amphetamine seems particularly harmful for females (Kogachi *et al*, 2017). Therefore, we tested whether amphetamine exerts a sex-specific effect on β cell development. We found that pancreatic insulin immunoreactivity was significantly reduced in female but not male neonates prenatally exposed to amphetamine (Fig 5A and A1). Glucagon levels, albeit reduced in offspring after prenatal amphetamine exposure, did not reach statistical significance (*but see* Fig EV4C and C1). When reconstructing neonatal pancreata by light-sheet microscopy, we find that it is not the number of islets *per se* [$55.3 \pm 11.9$ (saline) versus $72.3 \pm 17.2$ (amphetamine)] but rather their size and insulin immunoreactivity that seem reduced in prenatally amphetamine-exposed females (Fig 5B and B1 and Movies EV1 and EV2). It is noteworthy that both escitalopram (Fig 5A and A1) and genetic deletion of *Slc6a4* (Fig EV4C and C1, and Appendix Fig S3) phenocopied amphetamine effects in female offspring. Cumulatively, these data show that

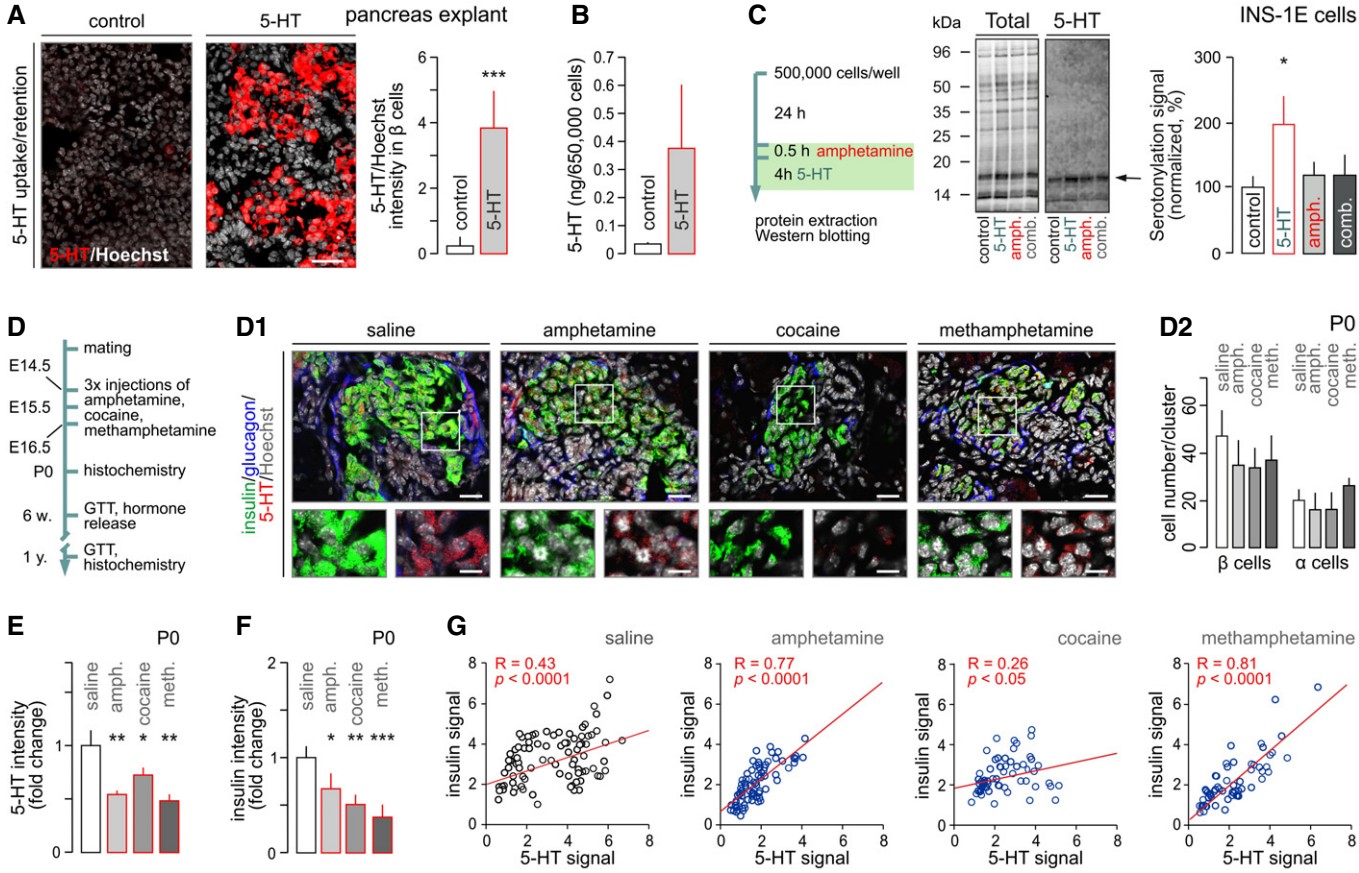

**Figure 4. Psychostimulants reduce protein phosphorylation *in vitro* and intracellular insulin and serotonin levels in pancreatic islets at birth.**

A    5-HT is taken up by pancreatic explants prepared from E13.5 mice. Data were expressed as means ± SEM. Experiments were performed in duplicate. *Scale bar* = 20 μm.

B    Likewise, INS-1E cells treated for 45 min with 5-HT (5 μM) accumulate this monoamine as measured by HPLC (means ± SEM; data are from duplicate experiments).

C    (*left*) Experimental outline to test amphetamine-induced protein serotonylation *in vitro*. (*middle*) Full-length Western blots showing total protein load upon Cy5-labeling and 5-HT-containing proteins. Arrow points to the protein band used for quantitative analysis (*right*). Note that amphetamine (10 μM) when combined with 5-HT (1 μM) eliminated the effect of the latter alone. Experiment was performed with quadruplicate biological samples. Data were expressed as means ± SEM.

D    Schema with methamphetamine (10 mg/kg), amphetamine (10 mg/kg), or cocaine (20 mg/kg) injected between E14.5 and 16.5. (D1) Representative images of neonatal tissues triple-labeled for insulin, glucagon, and 5-HT. Hoechst 33342 was used as a nuclear counterstain (pseudo-colored in gray). *Scale bars* = 15 μm and 6 μm (*inserts*). (D2) Prenatal exposure to psychostimulants does not affect either α or β cell numbers in the P0 mouse pancreas. Quantitative data from n ≥ 3 mice/group (from n = 3 pregnancies) were expressed as means ± SD.

E, F    *In utero* psychostimulant exposure significantly decreased 5-HT immunoreactivity (E). (F) Likewise, insulin immunoreactivity was reduced. Quantitative data from n ≥ 3 mice/group (from n = 3 pregnancies) were expressed as means ± SD.

G    Positive correlation between insulin and 5-HT immunosignals per β cell in control and after prenatal drug exposure. Pearson correlation was defined on data from n ≥ 60 cells/drug.

Data information: Data were analyzed by pair-wise comparisons after one-way ANOVA. ***$P < 0.001$, **$P < 0.01$, *$P < 0.05$.

pancreas development is sensitive to psychostimulant action in a sex-specific manner and uses SERT to disrupt insulin production by β cells.

## Prenatal exposure to psychostimulants impairs insulin expression and leads to glucose intolerance in adult offspring

If viewed as a developmental delay, the fetal deregulation of insulin production and signaling might be assumed as becoming compensated later in life. Alternatively, psychostimulant effects could persist throughout postnatal life and sensitize the offspring to developing glucose intolerance/diabetes. When analyzing 6-week-old offspring, we find that the number of β and α cells is not affected by prenatal drug exposure (Appendix Fig S4A). These data are compatible with our *ex vivo* observations (Fig 4D2), as well as continued β cell proliferation in postnatal pancreata (Taylor *et al*, 2015). Nevertheless, examination of sex effects in 6-week-old offspring showed profoundly reduced insulin immunoreactivity in females (*P* < 0.05 for all drugs versus saline; Fig 6A and A1) but not males (Fig 6A). At the same time, glucagon immunoreactivity remained unaffected

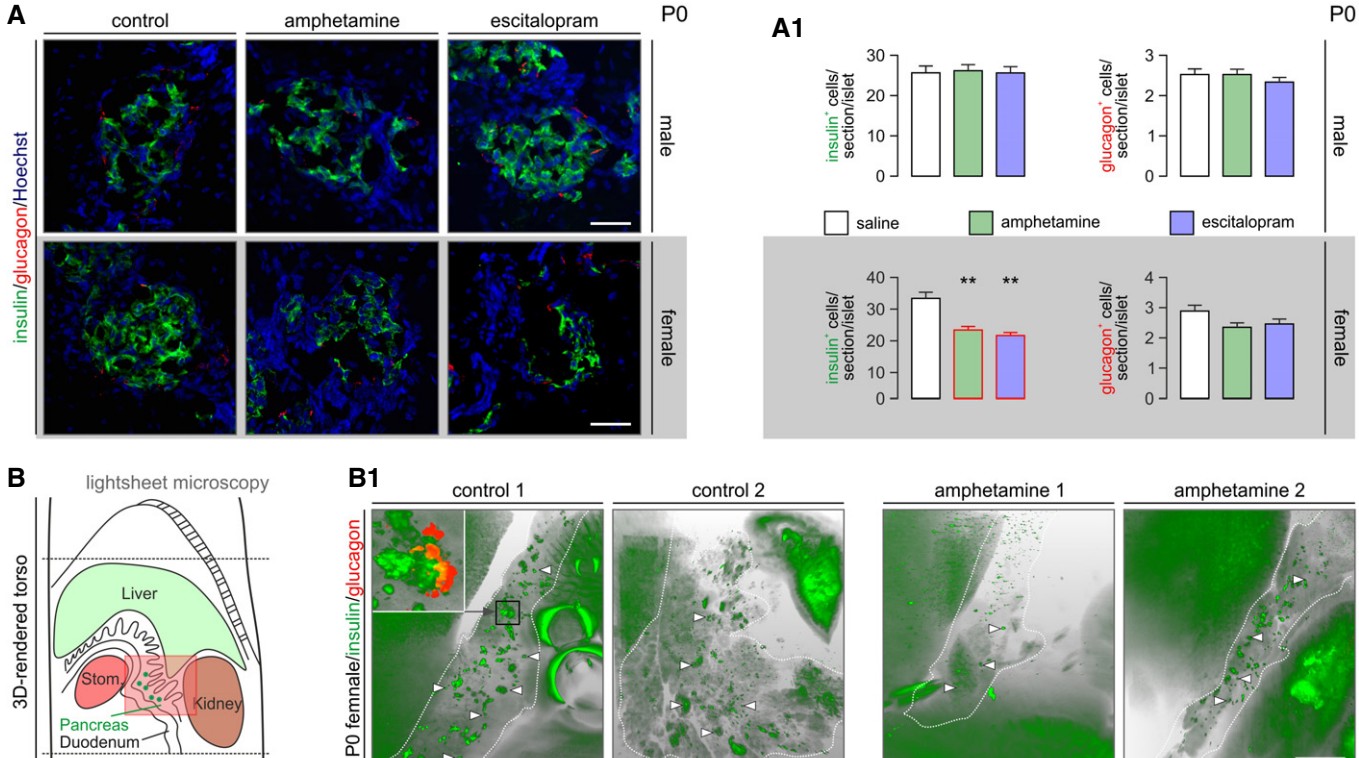

**Figure 5. Both amphetamine and escitalopram reduce insulin immunoreactivity in female offspring at birth.**

A  Histochemical examples of neonatal pancreata used for the simultaneous detection of insulin and glucagon. Hoechst 33342 was used as nuclear counterstain. *Scale bars* = 80 μm. (A1) Quantitative data from $n > 6$ mice/sex from independent pregnancies. Note that insulin immunoreactivity was significantly reduced in the pancreas of female but not male offspring. Data were expressed as means ± SEM, **$P < 0.01$ (pair-wise comparisons after one-way ANOVA). Figure EV4C and C1 is referred to data on $Slc6a4^{-/-}$ mice.

B  Longitudinal cross-sectional rendering of the neonatal mouse torso to aid pancreas visualization by light-sheet microscopy. The area with semi-transparent red overly is shown in B1. (B1) Insulin-labeled pancreata (encircled by *dashed lines*) from respective pairs of control and amphetamine-treated neonatal mice. Note the reduced size and labeling intensity for insulin of pancreatic islet-like structures (*arrowheads*) upon intrauterine amphetamine exposure. Inset reveals that the core–shell cytoarchitecture of an islet is retained in optically cleared intact tissues simultaneously processed for the detection of insulin and glucagon. Many peripheral organs showed significant autofluorescence. *Scale bar* = 1.5 mm. Videos of orthogonally reconstructed tissues are available as part of the Appendix file.

in both sexes (Appendix Fig S4B). These data were supported by reduced *Ins2* mRNA levels (Fig 6B).

We then asked whether the loss of insulin content affects glucose metabolism in psychostimulant-exposed offspring. Body weights of 6-week-old mice prenatally exposed to psychostimulants did not differ from controls (Appendix Fig S4C). Nevertheless, psychostimulant-exposed offspring were glucose intolerant: their blood glucose levels were significantly higher before and during glucose challenge for both females [$P < 0.05$ at 0, 30, 60, and 120 min (amphetamine) or (cocaine); $P < 0.05$ at 30 min (methamphetamine) versus (saline); Fig 6C] and males [$P < 0.05$ at 0, 30–120 min (amphetamine); $P < 0.05$ at 0, 60, 90, and 120 min (methamphetamine) versus (saline); Fig 6D]. Next, we plotted the "area under the curve" (AUC) to test for the relative severity of glucose intolerance, which revealed a more severe phenotype in females ($P < 0.05$ for all drugs; Fig 6E) than in males ($P < 0.05$ for amphetamine; Fig 6F).

Thereafter, we studied a 1-year-old cohort of mice prenatally exposed to amphetamine, which appeared as the most harmful drug at 6 weeks of age. Neither the size of (including β and α cells numbers; Appendix Fig S4D) nor the insulin content in

amphetamine-exposed pancreata was reduced relative to their respective controls in either sex, although amphetamine-exposed females tended to present lower insulin immunoreactivity (Fig 6G and G1). Female but not male mice prenatally exposed to amphetamine presented with reduced bodyweight ($P < 0.01$, Fig 6H). Remarkably, mice of both sexes with a history of amphetamine exposure showed hypoglycemia at rest (Fig 6I and J). In amphetamine-exposed females, the first phase of insulin release was normal. However, its second phase was significantly blunted (Fig 6I; *for AUC calculation see* Fig 6K). Amphetamine-exposed male mice showed blood glucose levels comparable to those of controls (Fig 6J and K). In sum, these data evidence a permanent deregulation of glucose homeostasis in adult offspring with a history of *in utero* psychostimulant exposure.

### *Pet1/Fev*-dependent deregulation of insulin expression as a candidate mechanism of psychostimulant action

The fact that episodic maternal intake of illicit psychostimulants could compromise the offsprings' glucose metabolism for life

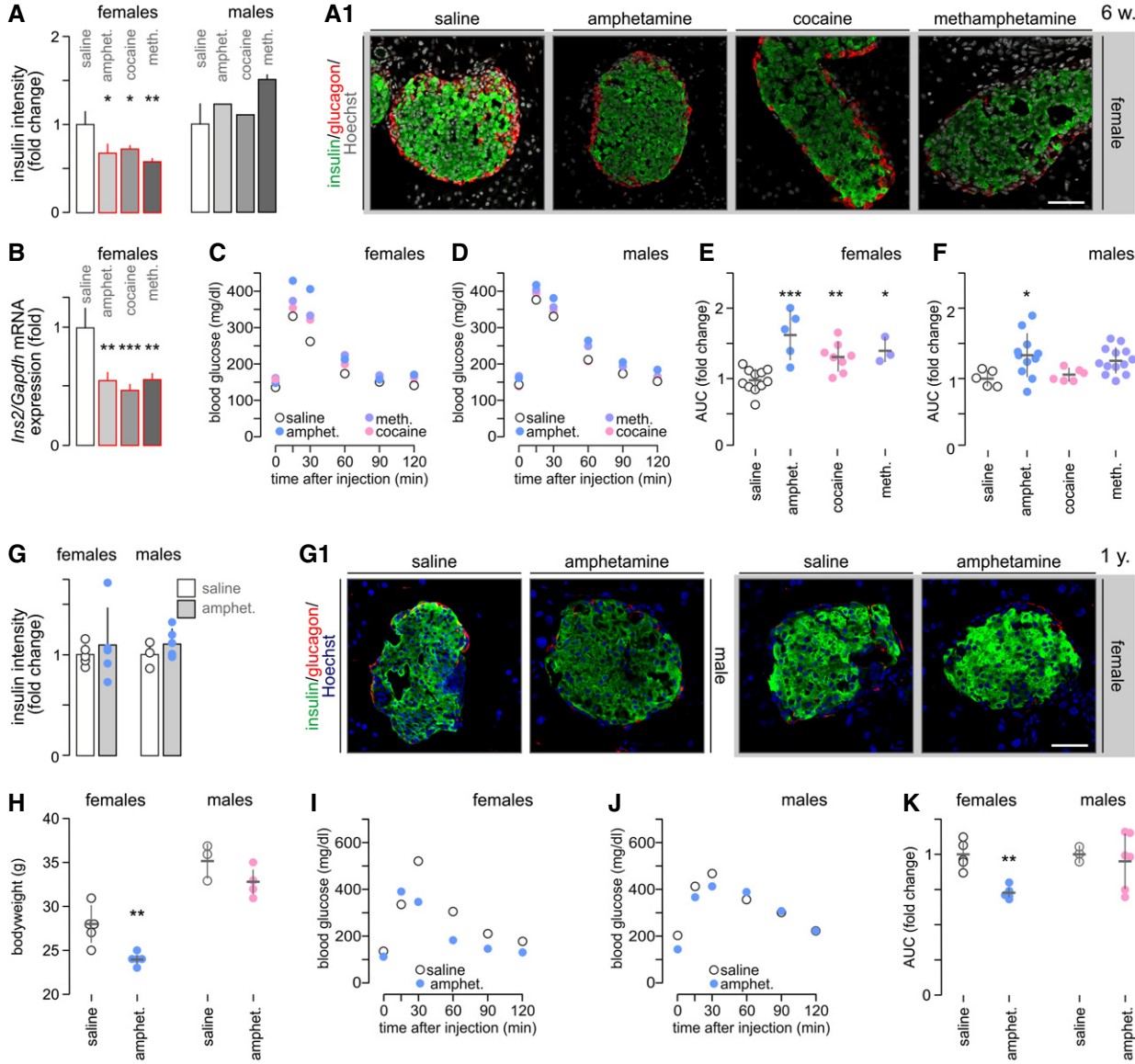

**Figure 6. Prenatal psychostimulant exposure impairs glucose homeostasis in adult offspring.**

A, B  Immunohistochemistry for insulin and glucagon in pancreatic islets of 6-week-old offspring born to drug-exposed mothers (A). Sex assignment shows that females respond to prenatal drug exposure with permanently reduced insulin levels. (A1) Representative images from females are shown and were counterstained with Hoechst 33342 (pseudo-colored in gray). *Scale bar* = 40 μm. (B) *Ins2* mRNA expression remains reduced in adult offspring exposed *in utero* to psychostimulants (pooled data). Quantitative data in (A, B) are from $n \geq 3$ mice/group and were expressed as means ± SD.

C, D  Blood glucose profiles during glucose tolerance test in females (C) and males (D) at 6 weeks of age.

E, F  Quantification of the "area under the curve" (AUC) indicates glucose intolerance in female mice prenatally exposed to psychostimulants with only amphetamine-treated males showing significantly increased AUC.

G  Insulin immunoreactivity no longer differed between amphetamine-exposed and control mice, irrespective of their sex, at 1 year of age. (G1) Representative images; *Scale bar* = 65 μm.

H  The bodyweight of 1-year-old female mice prenatally exposed to amphetamine was significantly reduced versus vehicle-treated controls.

I, J  Blood glucose profiles during glucose tolerance testing in females (I) and males (J) at 1 year of age.

K  AUC indicates glucose intolerance in female but not male mice prenatally exposed to amphetamine.

Data information: Quantitative data were normalized to those in control and expressed as means ± SD (for visual clarity SD were omitted in C, D, I, J); $n \geq 3$ animals/group/sex were analyzed (from 3 to 4 pregnancies each). ***$P < 0.001$, **$P < 0.01$, *$P < 0.05$ (pair-wise comparisons after one-way ANOVA).

prompted us to examine cellular determinants of adverse drug effects. Stimulation of 5-HT receptors on β cells has been shown to activate *Pet1/Fev*, an archetypal "5-HTergic transcription factor",

which promotes *Ins1/2* expression (Ohta *et al*, 2011). We hypothesized that psychostimulant-induced deregulation of 5-HT signaling might reduce *Pet1/Fev* expression, thus rendering it insufficient to

maintain insulin expression (Fig EV5A). In pancreatic explants, amphetamine ($P < 0.01$), cocaine ($P < 0.001$), and methamphetamine ($P < 0.001$) significantly reduced *Pet1/Fev* mRNA levels relative to controls (Fig EV5B). In contrast, 5-HT itself did not affect *Pet1/Fev* expression (Fig EV5B). Simultaneously, *Ins2* mRNA expression was reduced by both psychostimulants and 5-HT ($P < 0.01$ for all; Fig EV5B), suggesting a different mechanism of action for psychostimulants (SERT engagement) and 5-HT (receptor engagement). We then used a correlation analysis to link *Pet1/Fev* and *Ins2* expression at $R = 0.83$, $P < 0.001$ (Fig EV5B). At the same time, neither psychostimulants nor 5-HT affected glucagon (*Gcg*) expression significantly (Fig EV5C), which also failed to correlate with *Pet1/Fev* expression (Fig EV5C1).

Next, we probed if similar changes occur upon intrauterine psychostimulant exposure *in vivo*. In 6-week-old female mice, *Pet1/Fev* expression was reduced upon prenatal amphetamine and cocaine exposure (Fig EV5D) and exhibited a close correlation with *Ins1/2* mRNA levels (Fig EV5D). These data support our notion that psychostimulants impair a transcriptional checkpoint mandatory for insulin production in β cells.

Whether *PET1/FEV* might be relevant to insulin (*INS*) expression in humans [beyond the animal models described here and previously (Ohta *et al*, 2011)] is unknown. Therefore, we used bulk mRNA from pancreatic islets of humans stratified for glucose intolerance (Uhlen *et al*, 2005; Fadista *et al*, 2014), which shows that *PET1/FEV* expression indeed correlated with that of *INS* ($R = 0.41$, $P = 4.7e^{-9}$; Fig EV5E) and was differentially expressed between non-diabetic and type 2 diabetic donor islets (Appendix Fig S5A and A1). In contrast, a correlation between *PET1/FEV* and *GCG* expression did not reach statistical significance ($R = 0.14$, $P = 0.064$; Fig EV5E), suggesting that *PET1/FEV* in humans is also involved in determining physiological insulin availability. Interestingly, *SLC18A2* expression correlated positively with that of *INS* but not *GCG* in human pancreatic islets (Appendix Fig S6A–A3). In contrast, *SLC6A4* expression showed a negative correlation with *INS* but positive with *GCG* expression (Appendix Fig S6B–B3). These data are not unexpected since our single-cell RNA-seq analysis returned predominant expression of *SLC6A4* in human α cells (Fig EV1). Thus, our results substantiate the cell type-specific use of monoamines in human pancreatic islets.

### Epigenetic changes upon intrauterine amphetamine exposure

The finding that glucose intolerance persists postnatally in offspring exposed to psychoactive drugs *in utero* suggests significant changes in the epigenetic landscape of the amphetamine-treated pancreas. Considering the tight correlation of *Pet1/Fev* and *Ins* expression, one might assume that *Pet1/Fev* repression could directly severe insulin production. Alternatively, impairment of upstream regulatory gene networks or effectors at the level of, e.g., regulated exocytosis might account for lifelong glucose intolerance. To experimentally test these hypotheses, we first set up an *in vitro* assay of chromatin accessibility: INS-1E cells were exposed to amphetamine for 3 days followed by washout for 3 days (to discount acute drug effects), sample preparation and ATAC-seq (Fig 7A). This experimental design reduced *Ins1* mRNA expression (Fig 7A1). Our data showed that *Ins1*, *Ins2*, and *Pet1/Fev* are not subject to changes in chromatin accessibility upon amphetamine

treatment *in vitro* (Fig 7A2), and argue for a more indirect regulatory mechanism.

Next, we studied the DNA methylation landscape in P0 pancreata of mice of both sexes from amphetamine-treated and control dams (Fig 7B and B1). By using *methylKit* (Akalin *et al*, 2012), we performed independent tests with sex and amphetamine treatment as variables. We derived six gene sets for comparisons and visualized the lack of any of their intersections by *UpSetR* (Conway *et al*, 2017). This analysis returned similar results on base-pair and promoter levels for both pooled samples and upon overdispersion correction of covariates. Interestingly, the most prominent differences were revealed between amphetamine-treated females versus amphetamine-treated males, as well as between amphetamine-treated males versus control males. These data suggest that the epigenetic regulation of pancreatic cell identity and function is critically sex-dependent and that prenatal amphetamine exposure can significantly alter pancreatic DNA methylation. Our results also confirmed that *Ins1*, *Ins2*, and *Pet1/Fev* are not affected by either hyper- or hypomethylation (Fig 7C). Notably, we find hypomethylated states of critical regulatory regions (close to the transcription start site or promoter) in genes that specify pancreas development (*Lmx1b*, *Ism1*) are implicated upstream to *Pet1/Fev* and involved in 5-HT-related pathways (*Htr1a*; Fig 7C1) (Mellitzer *et al*, 2006; Ohta *et al*, 2011). Moreover, we defined sex- and amphetamine-dependent changes in DNA methylation for, e.g., *Gria1* (glutamate receptor subunit) and *Homer2* (postsynaptic scaffolding protein), which regulate sensitivity to glutamate (Fig 7C2). Even though the sample size of our analysis constitutes a potential confound, gene ontology (GO) enrichment analysis (Falcon & Gentleman, 2007) of genes for the "top 500" most perturbed promoters using amphetamine treatment as variable showed differences in an unexpectedly large number of pathways (Fig 7D), with 5-HT-related gene networks and regulatory gene sets most severely affected. Cumulatively, these data suggest that amphetamine permanently deregulates 5-HT signaling and affects regulatory gene networks hierarchically placed above *Pet1/Fev* and *Ins* to produce lifelong pancreas dysfunction.

## Discussion

Glucose homeostasis is maintained by hormone release from the endocrine pancreas. If this function is compromised, the body loses its ability to regulate blood glucose levels, which is causal to developing diabetes mellitus (both type 1 and 2) (Roder *et al*, 2016). Here, we show that prenatal exposure to the commonly abused psychostimulants amphetamine, cocaine, and methamphetamine leaves offsprings' glucose homeostasis permanently impaired, leading to glucose intolerance. Earlier, cytotoxic effects of psychostimulants were reported in the brain (Costa *et al*, 2013). However, we show that psychostimulants at doses known to induce behavioral effects in mice (Barenys *et al*, 2010; Lee *et al*, 2011; Khoradmehr *et al*, 2015) do not modify pancreatic islet cytoarchitecture *in utero*. Instead, these drugs impose functional modifications enduring into the adulthood of affected offspring. Our findings on reduced 5-HT signaling and *Pet1/Fev* expression are compatible with permanently decreased 5-HT levels in the brains of rats prenatally exposed to cocaine (Henderson & McMillen, 1993). However, we find that it is not *Pet1/Fev* itself whose expression is directly impacted by

psychostimulants. Instead, the many regulatory genes and archetypical transcription factors that drive pancreas specification (and particularly β cell differentiation) undergo long-term and sex-specific epigenetic modifications at the level of DNA methylation, thus being poised to impose lifelong expressional deregulation of insulin production.

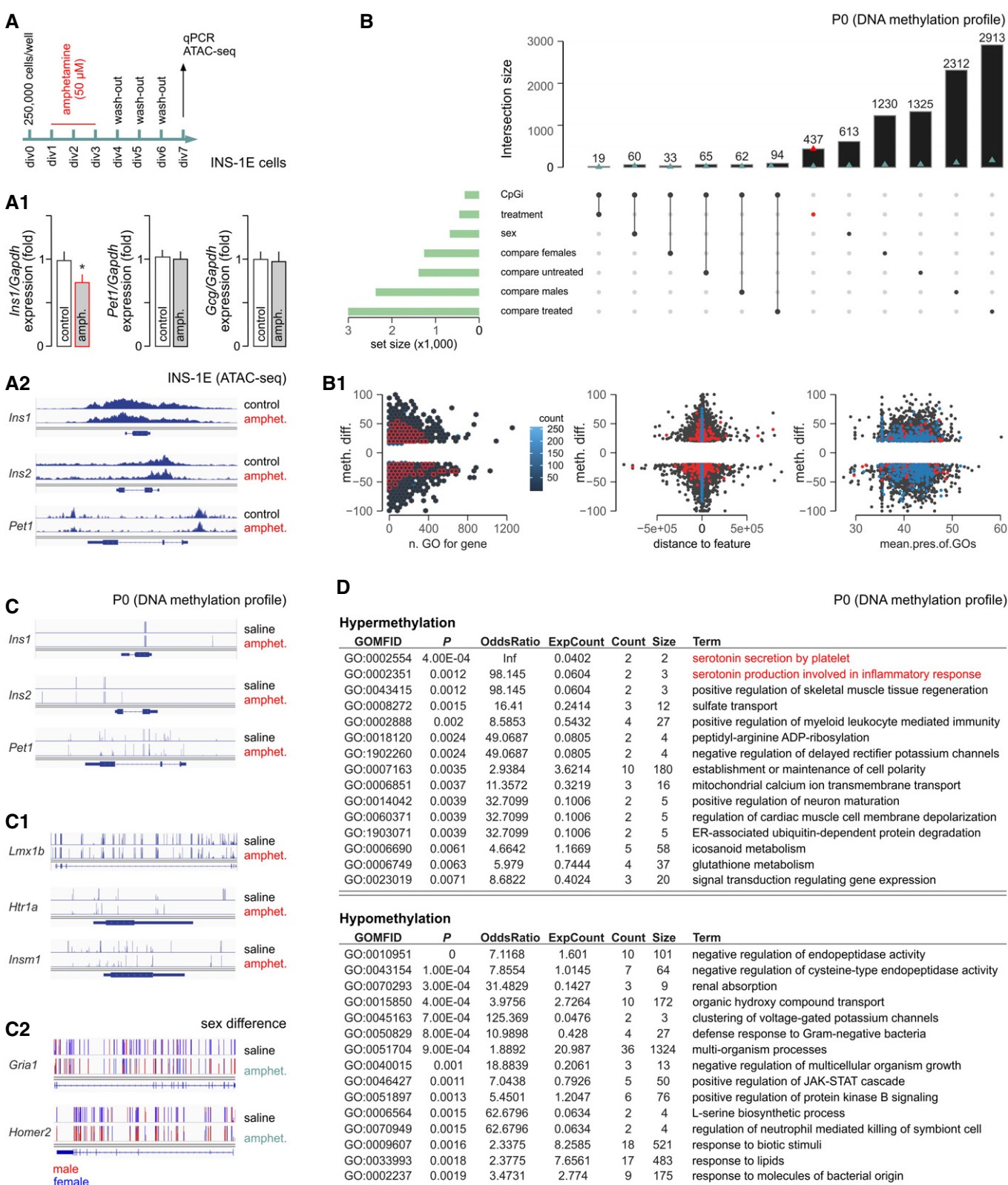

**Figure 7.**

◀

**Figure 7. Epigenetic changes upon amphetamine treatment *in vitro* and *in vivo*.**

A   Experimental design and (A1) expressional quality control (note reduced *Ins1* mNRA levels) in INS-1E cells treated with amphetamine. Data were expressed as means ± SD; *n* = 3; \**P* < 0.05 (Student's *t*-test). (A2) Genome browser plot showing ATAC-seq average signal intensity for *Ins1*, *Ins2*, and *Pet1/Fev* between control and amphetamine-treated INS-1E cells.

B   *UpSet* plot showing intersections of gene sets obtained independently by $\chi^2$-test with 0.01 Q-value cutoff. Blue triangles show events nearest to transcription start sites (TSSs), while red triangles highlight treatment-only-related difference on B (and B1) plots. (B1) The distribution of DNA methylation differences of locus (BP) to related variables (*left to right*: number of Gene Ontology (GO) terms for gene, distance to nearest TSS, mean presence of GO term (differentially methylated to the whole size of term ratio) relevant for gene-related locus) is shown.

C   Differences in DNA methylation in neonatal mice (*n* = 6 for control; *n* = 5 for amphetamine) for *Ins1*, *Ins2*, and *Pet1/Fev* (C), as well as *Lmx1b*, *Htr1a*, *Insm1* (central to 5-HT signaling; C1). (C2) Sex differences in DNA methylation for *Gria1* and *Homer2* (prototypic for glutamate signaling). Interactive genome browser tracks for all genes detected are available at the UCSC portal (http://bit.ly/AMPHpanDifMeth).

D   The most enriched GO terms associated with the top 500 hypermethylated (*top*) and hypomethylated (*bottom*) promoters of amphetamine-treated versus control mice based on the *RnBeads 2.0* pipeline (Muller *et al*, 2019). Red color indicates the pathways associated with 5-HT signaling. Complete datasets of differential methylation analysis for genes and promoters, as well as for sex and amphetamine effects, are available as Tables EV1–EV5.

The pharmacology of psychostimulant drugs is intensely studied (Volkow & Morales, 2015), particularly because of the many "legal highs" and synthetic compounds entering the recreational market (Korpi *et al*, 2015). Even though the biological half-life of amphetamine, methamphetamine (5–30 h), and cocaine (0.7–1.5 h) differs significantly, they affect intercellular communication by monoamines, with varying affinities for VMAT2 and cell-surface reuptake transporters (NET, DAT, and SERT) (Pifl *et al*, 1995; Kitanaka *et al*, 2006; Steinkellner *et al*, 2014; Ross *et al*, 2015; Sitte & Freissmuth, 2015). The powerful psychoactive effects of these drugs warrant a primary focus on the nervous system. Likewise, the pancreas is another legitimate target in view of its 5-HT-, DA-, and NE-mediated events, including paracrine communication between α and β cells, auto-regulatory feedback for hormone release, and the sympathetic control of hormonal output (Sorenson *et al*, 1979; Paulmann *et al*, 2009; Ustione & Piston, 2012; Borden *et al*, 2013; Almaca *et al*, 2016). Even if more indirectly, the appetite-suppressing effects of psychostimulants (Vicentic & Jones, 2007) also suggest a relationship between neural and endocrine centers defining food intake with the pancreas being central to the peripheral control of energy homeostasis.

By using RNA-seq, we find SERT (SLC6A4) but not NET (SLC6A2) or DAT (SLC6A3) expressed in human pancreatic β cells, which we confirmed in native mouse β and INS-1E cells. This finding was functionally substantiated by [$^3$H]5-HT uptake experiments. Even if human and mouse expression profiles correspond at many levels, we caution against the direct translation of the rodent data because critical differences between species exist in, e.g., VMAT2 expression (Schafer *et al*, 2013). When rationalizing our developmental biology results on 5-HT signaling, one ought to consider that: firstly, a "5-HT signaling cassette" (i.e., coincident expression of metabolic enzymes, VMAT2, 5-HT receptors, and SERT) exists in developing and neonatal β cells. This is reminiscent to the early neural organization of 5-HT signaling including the presence of SERT even before the emergence of serotonin neurons in the brain (by gestational weeks 8 and 15, respectively) (Jacobs & Azmitia, 1992). Moreover, a gradual increase in TPH expression in β cells during pregnancy (Hill, 2017) is critical to promote cell proliferation. Secondly, we show that 5-HT signaling, while functional in pancreatic primordia producing cell lineages during its secondary transition, can use a variety of 5-HT receptors. Our human fetal analysis demonstrated the expression of *HTR1B/1F*, *HTR2A/2B/2C*, and *HTR3E* mRNAs during the 1$^{st}$ trimester. These receptors could

be precisely assigned to α and β cells by single-cell RNA-seq, which distinguished *HTFR1F* (and perhaps *HTFR2B*) as predominant in human β cells. This receptor diversity was confirmed by histochemistry, is compatible with previous findings in mouse (Ohara-Imaizumi *et al*, 2013), and corroborates HTR1F-mediated β-to-α cell signaling in humans (Almaca *et al*, 2016), as well as Ca$^{2+}$-dependent auto-regulation of insulin release upon HTR2B or HTR3 activation (Ohara-Imaizumi *et al*, 2013). Thirdly, the availability of circulating 5-HT is regulated by both fetal and maternal 5-HT sources, the latter including transient placental 5-HT production (Bonnin *et al*, 2011) from tryptophan precursors (Hill, 2017). Therefore, we posit that the psychostimulant effects seen here could pivot on both indirect (due to the suppression of maternal food intake, subsequent glucose depletion, and placental deregulation) and direct (developmental reprogramming of pancreatic islets) components *in vivo*. Since psychostimulants cross the placenta (Thompson *et al*, 2009), suppress SERT expression and activity, and evoke permanent modifications to *Pet1/Fev* and *Ins* expression only in β cells (note that none of the parameters deviated significantly in α cells), we furnish the concept of direct reprogramming of the fetal pancreas by (meth-)amphetamine and cocaine.

Even with SERT being a solitary molecular target for psychoactive drugs, Ca$^{2+}$ imaging demonstrated that their acute application rapidly and significantly reduces glucose responsiveness in β cells. Each drug was tested in a broad range of concentrations and exhibited a dose–response relationship for the frequency of Ca$^{2+}$ oscillations. Particularly for 5-HT itself, the reduced frequency of Ca$^{2+}$ waves coincided with their increased amplitude, which we interpreted as a dynamic resetting and integration of subthreshold Ca$^{2+}$ transients to produce enlarged responses. Given that Ca$^{2+}$ spikes in β cells are considered as indices of primed exocytosis, we entertain the possibility that modulating SERT in β cells could powerfully change the temporal dynamics of insulin release. More broadly, the pathophysiological sign of amphetamine, methamphetamine, and cocaine use during pregnancy was uniform with reduced insulin content in β cells. The fact that escitalopram, a selective SERT inhibitor, as well as genetic *Slc6a4* ablation in mice reminisced amphetamine effects suggests a direct SERT involvement rather than, e.g., the allosteric modulation of 5-HTR$_{1A}$ or 5-HTR$_{1B}$ (Borycz *et al*, 2008). This is also conceivable for our *in vitro* studies in which bath superfusion of the drugs could have limited the buildup of extracellular 5-HT and rendered the engagement of 5-HT auto-receptor mechanisms less prominent.

*In utero* psychostimulant exposure rendered hormonal output from the endocrine pancreas deficient in affected offspring. This is not unexpected with experimental evidence on epigenetic modifications mounting for the most common illicit drugs (Dinieri *et al*, 2011; Robison & Nestler, 2011). Particularly for maternal (meth-) amphetamine abuse during pregnancy, female rather than male offspring exhibit severe deficits in both people (Kogachi *et al*, 2017) and animal models (Castner & Becker, 1996). Histone serotonylation emerges as a novel epigenetic mechanism, which was shown to limit cell differentiation (Farrelly *et al*, 2019). Thus, fetal deregulation of intracellular 5-HT availability by psychostimulants could contribute to the developmental delay for β cells, which is supported by the reduction of *NeuroD1* and *Nkx6.1* transcription factors in our animal models. Indeed, we found long-lived intracellular 5-HT accumulation upon 5-HT loading of both pancreatic explants and INS-1E cells, which was occluded by amphetamine. A plausible explanation for this phenomenon is that 5-HT became covalently bound to glutamine residues of proteins in a reaction catalyzed by transglutaminases. A lifelong change in SERT-mediated 5-HT loading could become of pathological significance because residual 5-HT executes rate-limiting feedback for insulin secretion by serotonylation of, e.g., *Rab3a* and *Rab27a* GTPases (Paulmann *et al*, 2009), critical for SNARE assembly and the priming of vesicle docking. Considering that DNA hypermethylation was most common amongst genes assigned to 5-HT release by GO pathway analysis, and 5-HT is co-released with insulin from mature β cells (Suckale & Solimena, 2010), we favor the hypothesis that reduced protein serotonylation links adverse psychostimulant effects to epigenetic modifications persisting throughout life in β cells of affected offspring.

In sum, we identify a pathological principle by linking maternal psychostimulant abuse during pregnancy to impaired glucose homeostasis throughout life. Considering the continued rise in the consumption of illicit drugs by young adults, including women of child-bearing age, we warn against gestational exposure to "5-HT modifying drugs", such as amphetamines and cocaine, to minimize harm to the unborn fetus.

# Materials and Methods

### Gene expression analysis in human subjects

Tissue biopsies from terminated foetuses (7–14 gestational weeks) were used to describe mRNA expression patterns for monoamine signaling systems in fetal pancreata (Fig 1). DNA and RNA were extracted using the TRIZOL method (Life Technologies) with quality control performed on an Agilent 2200 Tapestation system (Bio-Rad) and quantification using Qubit. RNA libraries were then constructed using the TruSeq RNA library preparation kit (Illumina). RNA sequencing was performed on a HiSeq 2000 system (Illumina). Paired-end 101 bp-long reads were aligned to the Reference Human Genome Build 37 using STAR. Gene expression was measured as the normalized sum of expression of all exons. The "*dexseq_count*" python script (http://bioconductor.org/packages/DexSeq/) was used to count uniquely mapped reads in each exon. Gene and exon expression normalizations were then performed using the TMM method in edgeR.

Gene–type 2 diabetes relationships were studied in islets from 202 cadaver donors of European ancestry (Nordic Islet Transplantation Programme; http://www.nordicislets.org). All procedures were approved by the ethics committee at Lund University and performed as described (Fadista *et al*, 2014). Briefly, purity of the islets was assessed by dithizone followed by islets culture in CMRL 1066 (ICN Biomedicals) supplemented with 10 mM HEPES, 2 mM L-glutamine, 50 μg/ml gentamicin, 0.25 μg/ml fungizone (Gibco), 20 μg/ml ciprofloxacin (Bayer Healthcare), and 10 mM nicotinamide at 37°C (5% $CO_2$) for 1–9 days prior to RNA preparation. Total RNA was isolated with the AllPrep DNA/RNA Mini Kit (Qiagen). RNA quality and concentration were measured using an Agilent 2100 bioanalyzer (Bio-Rad) and a Nanodrop ND-1000 (NanoDrop Technologies), respectively. Samples were stratified based on glucose tolerance estimated from HbA1c, i.e., donors with normal glucose tolerance (HbA1c < 6%, $n = 123$), impaired glucose tolerance (IGT, 6% ≤ HbA1c < 6.5%, $n = 47$), and type 2 diabetes (HbA1c ≥ 6.5%, $n = 32$). A linear model adjusting for age and sex as implemented in the R Matrix eQTL package was used to determine the expression of genes associated with glucose tolerance status. Raw data were base called and de-multiplexed using CASAVA 1.8.2 (Illumina) before alignment to hg19 with STAR. To count the number of reads aligned to specific transcripts, featureCounts (version 1.4.4, http://bioinf.wehi.edu.au/featureCounts/) was used. Raw data were normalized using trimmed mean of M-values (TMM) and transformed into $log_2$ counts per million ($log_2$CPM) using voom (limma R, Bioconductor) before linear modeling. After TMM normalization, batch effects were removed using the Combat function (sva package).

### Single-cell RNA-seq of pancreatic islets

Analysis was performed based on a publicly available expression matrix (Segerstolpe *et al*, 2016) from 270 human β cells and 874 human α cells from 6 healthy and 4 type 2 diabetic donors, which was originally generated on human tissue and primary islets from Prodo Laboratories Inc., providing islets isolated from donor pancreata obtained from deceased individuals with research consent from Organ Procurement Organizations. Single-cell RNA-seq libraries were produced with the Smart-seq2 protocol (Segerstolpe *et al*, 2016). Sequencing was carried out on an Illumina HiSeq 2000 generating 43 bp single-end reads. Sequence reads were aligned toward the human genome (hg19 assembly) using STAR (v2.3.0e), and uniquely aligned reads within RefSeq gene annotations were used to quantify gene expression as RPKMs using the "rpkmforgenes" routine (http://sandberg.cmb.ki.se/rnaseq/).

### Mice

On embryonic days (E) 14.5, 15.5, and 16.5, pregnant C57BL/6N mice (Charles River) were intraperitoneally injected with amphetamine ($n > 10$; 10 mg/kg body weight), methamphetamine ($n = 6$; 10 mg/kg body weight), cocaine ($n = 6$, 20 mg/kg body weight), or saline ($n > 10$; as control) at an injection volume of 10 ml/kg. All drugs were obtained from Sigma. Wild-type (*Slc6a4*$^{+/+}$) and phenotypically *Slc6a4* null fetuses on a C57BL/6J background were generated by mating heterozygous *Slc6a4*-Cre$^{+/-}$ knock-in mice (B6.129-Slc6a4$^{tm1(cre)Xz}$/Cnrm; strain EM:04890 from the European

Mouse Mutant Archive) with $Cre^{+/+}$ predicted to completely disrupt *Slc6a4* function ($n = 4$ dams; Appendix Fig S3). Only wild-type and null littermate offspring were analyzed after daily *in utero* exposure to amphetamine (E14.5–E16.5). On P0, offspring in each litter was randomly chosen and sacrificed for analysis. This was done such that each litter was kept equal in size until weaning. Animals were kept in groups of 6 under standard housing conditions with a 12-h/12-h light/dark cycle and food and water available *ad libitum*. Experiments on live animals conformed to the 2010/63/EU European Communities Council Directive and were approved by the Austrian Ministry of Science and Research (66.009/0145-WF/II/3b/2014, and 66.009/0277-WF/V/3b/2017). Particular effort was directed toward minimizing the number of animals used and their suffering during experiments.

### GTT assay

Six-week-old offspring prenatally treated with amphetamine (five females; 11 males), methamphetamine (three females; 13 males), cocaine (eight females; six males), and saline (11 females; five males) from 3 to 4 pregnancies/drug, as well as 1-year-old mice exposed to amphetamine (six females; five males) or saline (five females; three males) *in utero* were analyzed. Animals were fasted overnight (~15 h) with water available *ad libitum*. Blood glucose levels were measured from tail blood using a FreeStyle Lite glucose meter (Abbott Diabetes Care) before (baseline, 0 min), as well as 15, 30, 60, 90, and 120 min after *i.p.* glucose challenge (2 g/kg). Data were expressed as mg/dl blood glucose and analyzed as "area under curve (AUC)", a descriptor representative of overall change during the sampling period.

### Isolation of pancreatic islets

Islets were obtained after perfusion of the pancreas with Hank's balanced salt solution (HBSS; Invitrogen/Gibco) containing collagenase (type I, 0.33 mg/ml; Sigma) and HEPES (25 mM), followed by purification on Histopaque 1,077 gradients (Sigma) (Szot *et al*, 2007). After repeated washes in HBSS containing 10% fetal bovine serum (FBS), isolated islets were maintained in RPMI-1640 medium supplemented as above in humidified atmosphere (5% $CO_2$) at 37°C overnight prior to measuring hormone release.

### Organotypic cultures of embryonic pancreata

Dorsal pancreatic buds were dissected on E13.5 under a stereoscope and cultured on 0.4 µm pore-sized filters (Millicell-CM; Millipore) in DMEM (Gibco) supplemented with N2 (Gibco) and streptomycin–penicillin–glutamine (Gibco) (Serafimidis *et al*, 2017). Equivalent to E14.5, E15.5, and E16.5 (i.e., 1–3 days *in vitro* (DIV)), explants were transferred to fresh medium that had optionally been supplemented with amphetamine (10 µM), cocaine (10 µM), methamphetamine (10 µM), or 5-HT (0.5 µM; Tocris). On DIV4, all explants were transferred to fresh medium and cultured for 2 more days. Culture media were then collected with explants either fixed for immunohistochemical analysis or snap frozen for gene expression profiling. Quantitative analysis was performed using ImageJ1.45 with appropriate plug-ins on $n = 4$–6 explants/group.

Table 1. Primer sequences used for quantitative analysis in this study.

| Gene | Primer sequence | Species |
|------|-----------------|---------|
| *Gapdh* | Forward: 5′-AACTTTGGCATTGTG-3′ | Mouse |
|  | Reverse: 5′-ACACATTGGTAG-3′ |  |
| *Gcg* | Forward: 5′-TACACCTGTTCGCAGCTCAG-3′ | Mouse |
|  | Reverse: 5′-CTCTGTGTCTTGAAGGGCGT-3′ |  |
| *Gcg* | Forward: 5′-AGGAACCGGAACAACATTGC-3′ | Rat |
|  | Reverse: 5′-CCTTTCACCAGCCAAGCAAT-3′ |  |
| *Htr2b* | Forward: 5′-TGTTTGTGATGCCGATTGCC-3′ | Mouse |
|  | Reverse: 5′-AGTTGCACTGATTGGCCTGA-3′ |  |
| *Ins1* | Forward: 5′-CCACCCAGGCTTTTGTCAAA-3′ | Mouse |
|  | Reverse: 5′-CCAGTTGTTCCACTTGTGGG-3′ |  |
| *Ins1* | Forward: 5′-GTGGGGAACGTGGTTTCTTC-3′ | Rat |
|  | Reverse: 5′-TGCAGCACTGATCCACAATG-3′ |  |
| *Ins2* | Forward: 5′-CCACAAGTGGCACAACTGGA-3′ | Mouse |
|  | Reverse: 5′-CTACAATGCCACGCTTCTGC-3′ |  |
| *Ins2* | Forward: 5′-GTGACCAGCTACAGTCGGAA-3′ | Rat |
|  | Reverse: 5′-GAGCTTCCACCAAGTGAGAAC-3′ |  |
| *NeuroD1* | Forward: 5′-GCTCCAGGGTTATGAGATCGT-3′ | Mouse |
|  | Reverse: 5′-CATCTGTCCAGCTTGGGGGA-3′ |  |
| *Nkx6-1* | Forward: 5′-GCACGCTTGGCCTATTCTCT-3′ | Mouse |
|  | Reverse: 5′-GAAGTCCCCTTGAGCCTCTC-3′ |  |
| *Pet1/Fev* | Forward: 5′-AGATCCCGTCGGAGATGGTC-3′ | Mouse |
|  | Reverse: 5′-GGTCGGTGAGCTTGAACTCG-3′ |  |
| *Pdx1* | Forward: 5′-ACTTAACCTAGGCGTCGCAC-3′ | Mouse |
|  | Reverse: 5′-AGCTCAGGGCTGTTTTTCCA-3′ |  |
| *Slc6a4* | Forward: 5′-GGCGCGAGGGTCGAG-3′ | Mouse |
|  | Reverse: 5′-ATGCTGGTCAGTCAGTG-3′ |  |
| *Tbp* | Forward: 5′-ACCCCTTCACCAATG ACTCCTATG-3′ | Mouse |
|  | Reverse: 5′-ATGACTGCAGCAAATCGCTTGG-3′ |  |
| *Tbp* | Forward: 5′-ACAGCCTTCCACCTTATGCT-3′ | Rat |
|  | Reverse: 5′- GCTGCTGTCTTTGTTGCTCT-3′ |  |
| *Tph2* | Forward: 5′-TCTACACCCCGGAACCAGAT-3′ | Mouse |
|  | Reverse: 5′-GCAAAGGCCGAACTCGATTG-3′ |  |

qPCR products were performed with primer pairs amplifying short fragments for each gene. Primer pairs were custom-designed to efficiently anneal to homologous nucleotide sequences from mouse.

### RNA isolation and gene expression analysis

Total RNA from P0 pancreata, as well as adult pancreatic islets, was isolated using the Aurum Total RNA Mini Kit (Bio-Rad) followed by DNase digestion. cDNA was prepared by reverse transcription with random primers using a High-capacity cDNA Reverse Transcription Kit (Applied Biosystems). Quantitative PCR analysis (CFX Connect, Bio-Rad) was performed using 50 ng of cDNA template, iTaq Universal SYBR Green Supermix (Bio-Rad), and specific primer pairs (Table 1).

## HEK293 cells, transfection, and maintenance

Human embryonic kidney (HEK) 293 cells were grown in minimum essential medium with Earle's salts and L-glutamine, 10% heat-inactivated FBS, and gentamicin (50 mg/l; all from Sigma). Cells were typically grown in 60- or 100-mm tissue culture plates (polystyrene, Falcon) at 37°C in 5% $CO_2$ atmosphere. The human SERT (SLC6A4) was stably expressed using a pRc/CMV vector (Thermo Fisher; Ca-phosphate transfection) followed by selection with G418 (1 mg/ml) as described (Pifl et al, 1996).

## Tissue preparation and immunohistochemistry

Adult mice were transcardially perfused with ice-cold 0.1 M Na-phosphate buffer (PB), followed by 4% paraformaldehyde (PFA) in PB (100 ml at 3 ml/min flow speed). Pancreata were rapidly dissected and post-fixed in 4% PFA overnight. P0 mice were sacrificed with their torsos immersion fixed in 4% PFA overnight. After equilibrating in 30% sucrose for 48–72 h, tissues were cryosectioned at either 14 μm (adult and P0) or 8 μm (E14.5) thickness (Leica CM1850) and thaw-mounted onto SuperFrost$^+$ glass slides. After rinsing in 0.1 M PB, specimens were exposed to a blocking solution composed of 0.1 M PB, 10% normal donkey serum, 5% BSA, and 0.3% Triton X-100 for 3-h followed by 48-h incubation with select combinations of primary antibodies (Table 2). Carbocyanine (Cy)2,3 or 5-conjugated secondary antibodies (1:300, Jackson ImmunoResearch) were applied in 0.1M PB supplemented with 2% BSA (20–22°C, 2 h). Nuclei were routinely counterstained with Hoechst 33342 (1:10,000; Sigma). Histochemical detection of Pdx1 required antigen retrieval, e.g., incubating glass-mounted sections in 1× target retrieval solution (pH 6.0, Dako) at 95°C for 15 min followed by repeated washes in 0.1 M PB. Sections were coverslipped using Fluoromount aqueous mounting medium (Sigma). Tissues were photographed on a Zeiss LSM880 laser-scanning microscope equipped with a Plan-Apochromat 20×/air objective at 1.0–2.5× optical zoom. Images were acquired in the ZEN2010 software package. Quantitative analysis was performed using ImageJ1.45 with appropriate plug-ins on $n \geq 3$/group with $n = 8$–10 islets per animal analyzed. Multi-panel images were assembled in CorelDraw X7 (Corel Corp.).

## Optical tissue clearing and light-sheet microscopy

Pancreata were excised en bloc, repeatedly washed in PB, and optically cleared by immersion in "CUBIC reagent 1" (25% urea, 25% N,N,N',N'-tetrakis(2-hydroxypropyl)ethylenediamine and 15% Triton X-100) for 1 week (Susaki et al, 2014). Insulin immunohistochemistry was performed by first blocking the specimens in 2% BSA, 5% NDS in 0.1M PB for 3–6 h, and then exposing the specimens to rabbit anti-insulin antibody (1:450; Table 2) in 0.1% BSA, 2% NDS, 5% DMSO, and 0.3% Triton X-100 on an orbital shaker at 37°C for 7 days. This incubation step was followed by repeated rinses in 0.1M PB with subsequent exposure to Cy2-conjugated secondary antibody raised in rabbit (1:400; Jackson) at 37°C for 5 days. Samples were then rewashed in PB and submerged in "CUBIC reagent 2" (50% sucrose, 25% urea, 10% 2,20,20′-nitrilo-triethanol, and 0.1% Triton X-100) for further clearing (additional

**Table 2. List of antibodies used in this study.**

| Antibody | Source | Concentration | Supplier & Cat. No. |
|---|---|---|---|
| 5-HT | Rabbit | 1:100 (IHC) | Dr. T.F. Görcs[a] |
|  |  | 1:400 (WB) |  |
| 5-HTR1A | Rabbit | 1:65 (IHC) | Atlas/Sigma, #HPA018073 |
| 5-HTR1F | Rabbit | 1:50 (IHC) | Atlas/Sigma, #HPA005555 |
| B-Actin | Mouse | 1:2,000 (WB) | US Biological, #A0760-40 |
| Glucagon | Mouse | 1:2,000 (IHC) | Sigma, #G2654 |
| Insulin | Guinea pig | 1:500 (IHC) | DAKO, #A0564 |
|  | Mouse | 1:500 | Cell Signaling, mAb#8138 |
|  | Rabbit | 1:450 | Cell Signaling, mAb#3014 |
| PDX1 | Rabbit | 1:500 | Cell Signaling, mAb#5679 |
| SERT | Guinea pig | 1:100 (IHC) | Chemicon, #AB1772 |
|  |  | 1:500 (WB) | Santa Cruz, #sc-1458 |
| TPH2 | Rabbit | 1:100 | Atlas/Sigma, #HPA046274 |

5-HT, 5-hydroxytryptamine (serotonin); 5-HTR1A, 5-hydroxytryptamine receptor 1A; 5-HTR1F, 5-hydroxytryptamine receptor 1F; PDX1, pancreatic and duodenal homeobox 1; SERT; Serotonin transporter; TPH2, Tryptophan hydroxylase 2.
Commercially available primary antibodies (except for 5-HT) were used for immunohistochemistry. Vendors and catalogue numbers (Cat. No.) are given for reasons of reproducibility.
[a]Dr. Tamás F. Görcs (formerly of Semmelweis University, Budapest Hungary) passed away on 14.01.2014.

5–8 days). Cleared tissues were imaged in "CUBIC reagent 2" with a measured refractory index of 1.45 orthogonal on a Lightsheet Z.1 (Zeiss) microscope using a 5× detection objective, 5× illumination optics, and laser excitation at 488 nm. Each plane was illuminated from a single side with images captured at 0.9× optical zoom at 3.5-μm z-stack intervals and an exposure time of 150 ms. Three-dimensional-rendered images were visualized with Arivis Vision4D for Zeiss (v. 2.12). Brightness and contrast of the images were manually adjusted to aid visual clarity. Quantitative analysis was performed in Arivis. Optionally, tissue clearing involved the simultaneous detection of insulin (1:500; Dako) and glucagon, with mouse anti-glucagon (1:500; Sigma) and Cy3-conjugated donkey anti-mouse used as secondary antibody (1:400; Jackson) for the latter.

## High-performance liquid chromatography

INS-1E cells were seeded at a density of 650,000 cells/well in 6-well plates and serum starved overnight. Subsequently, cells were exposed to 5-HT (5 μM) for 45 min (in duplicate). After washing in ice-cold 0.05 M phosphate-buffered saline (2×), samples were scraped into 300 μl of perchloric acid and Na-bisulfite (final concentrations are 0.1 M and 0.4 mM, respectively), ultrasonicated, and centrifuged at 16,100 g for 10 min. For the determination of intracellular 5-HT levels, 100 μl supernatant was injected directly into a HPLC system equipped with a LiChroCART® 250-4 column (RP18μ, 5 μm, VWR) and a BAS electrochemical detector (Bioanalytical System). The mobile phase consisted of 0.1 M sodium acetate buffer (pH 4.3) containing 1 mM EDTA, 0.2 mM l-heptane sulfonic acid, 0.1% triethylamine, 0.2% tetrahydrofuran, and 10% methanol.

## Radiolabeled ligand uptake and associated pharmacology

INS-1E cells were seeded at a density of 250,000 cells/well in 24-well plates and incubated with 0.3 μM [$^3$H]5-HT, [$^3$H]DA, or [$^3$H]NE (all from Perkin Elmer) in uptake buffer with/without escitalopram (10 μM; SERT inhibitor) or mazindol (10 μM, Novartis; dual DAT/NET inhibitor) at 37°C for 5 min. To test whether VMAT2 is operational and contributes to amphetamine effects in INS-1E cells, we exposed samples (250,000 cells/well grown for 3 days *in vitro*) to 0.1 μM [$^3$H]5-HT in uptake buffer containing 3, 6, or 15 mM D-glucose in the absence or presence of reserpine (300 nM, Sigma; a VMAT2 inhibitor) or escitalopram (10 μM, Sigma; SERT inhibitor) at 37°C for 45 min. Intracellular [$^3$H] was measured in a scintillation counter (Beckman) and considered as a 1:1 indicator of catecholamine content.

## Ca$^{2+}$ imaging

Ca$^{2+}$ responses of INS-1E cells were recorded using Fura-2AM (Invitrogen). Ratiometric imaging (Malenczyk *et al*, 2017) was performed by using a VisiChrome monochromator (Visitron Systems), and CoolSnap HQ$^2$ back-cooled camera (Photometrics) mounted onto a Zeiss Axiovert microscope equipped with a water-immersion 20×/differential interference contrast objective (Plan-Apochromat/N.A. 1.0). Ca$^{2+}$ responses were recorded using the VisiView software (Visitron Systems) and evaluated in GraphPad Prism 7 (GraphPad). Measurements were performed in standard Krebs-Ringer basic external solution containing (in mM): 119 NaCl, 2.5 KCl, 1 NaH$_2$PO$_4$, 1.5 CaCl$_2$, and 1.5 MgCl$_2$, 20 HEPES, and "high" (11) or "low (3) glucose (pH 7.4). Drugs were directly applied into the recording chamber at the final concentrations indicated. Experiments were performed at 25°C.

## In vitro electrophysiology

HEK293 cells at a density of 1.5 × 10$^4$ were split onto poly-D-lysine-coated glass coverslips (12 mm in diameter) 48 h prior to electrophysiological recordings. HEK293 cells were kept in Krebs-Ringer solution that served as external solution with a composition (in mM): 119 NaCl, 2.5 KCl, 1 NaH$_2$PO$_4$, 1.5 CaCl$_2$, and 1.5 MgCl$_2$, 20 HEPES, 11 glucose (pH 7.4). For whole-cell recordings, patch pipettes were pulled from borosilicate glass with a programmable Flaming-Brown micropipette puller (P-100; Sutter Instruments), heat-polished to a final tip resistance of 4–5 MΩ, and filled with (in mM): 140 KCl, 2 MgATP, 0.5 EGTA, and 10 HEPES (pH 7.4 was adjusted with kOH). Whole-cell currents were recorded by clamping the cells at −70 mV holding potential and continuously superfused with either bath or drug-containing solutions (25 μM amphetamine, 10 μM cocaine, 5 μM methamphetamine, or 10 μM escitalopram) through a peristaltic pump at a flow rate of 2 ml/min. Recordings were made on an EPC-10 triple amplifier controlled by PatchMaster 2.80 (HEKA) at 25°C. Traces were filtered and analyzed off-line using ClampFit (Axon Instruments).

## Quantitative Western blotting with total protein normalization

Total protein labeling was initiated by adding Cy5 dye reagent (GE Healthcare) that had been pre-diluted (1:10) in ultrapure water.

Samples were mixed and incubated for 5 min at 21–24°C. The labeling reaction was terminated by adding Amersham WB loading buffer (GE Healthcare; 20 μl/sample) containing 40 μM DTT. Samples were then boiled at 95°C for 3 min with equal amounts (20 μg/40 μl) and subsequently loaded onto an Amersham WB gel card (13.5%). Electrophoresis (600 V, 42 min) and protein transfer onto polyvinylidene-difluoride membranes (100 V, 30 min) were at default settings in an integrated Amersham WB system (GE Healthcare) for quantitative SDS–PAGE and Western blotting of proteins with fluorescence detection. After blocking, membranes were incubated with rabbit anti-5HT antibody (1:400, kindly provided by T.J. Görcs) overnight. Antibody binding was detected by using species-specific (anti-rabbit) Cy3-labeled secondary antibodies (1:1,000; GE Healthcare). Membranes were dried before scanning at 560 nm excitation. Automated image analysis was performed with the Amersham WB evaluation software with manual optimization if necessary.

## Chromatin accessibility by ATAC-seq

Open chromatin mapping was performed with the assay for transposase accessible chromatin (ATAC-seq) (Buenrostro *et al*, 2015) with minor modifications (Rendeiro *et al*, 2016). In each experiment, 1 × 10$^5$ INS-1E cells were incubated in the transposase reaction mix (12.5 μl 2× TD buffer, 2 μl T$_N$5 transposase (Illumina), 0.1% NP-40, and 10.25 μl nuclease-free water) at 37°C for 30 min. After DNA purification with the MinElute kit (Qiagen), 1 μl of the eluted DNA was used in a qPCR product to estimate the optimum number of amplification cycles. Library amplification was followed by SPRI size selection to exclude fragments larger than 1,200 bp. DNA concentration was measured with a Qubit fluorometer (Life Technologies). Sequencing was performed on an Illumina HiSeq 3,000/4,000 instrument in single-end 50 bp mode. Raw sequencing data were processed by *atacseq* NextFlow pipeline (Dockerfile v.1.0.0) to perform library-level quality control and analysis. Merged bigWigs were normalized, peaks called, consensus sets created, counted, and processes for differential accessibility. Interactive genome browser tracks are available at the UCSC portal (http://bit.ly/AMPHpanDifMeth).

## DNA and RNA extraction from frozen pancreas tissues

Pancreas samples were homogenized by using a TissueLyser II (Qiagen) with the following settings: 4 × 30 s, 30 Hz with RNAse/DNAse-free ceramic beads. Tissues were dissolved in RLT lysis buffer of the DNA/RNA Allprep Kit (Qiagen) following the manufacturer's instructions. DNA quality (fragmentation) was checked on an 0.8% agarose gel.

## DNA methylation

Extracted DNA was subjected to the reduced representation bisulfite sequencing (RRBS) workflow as described previously (Klughammer *et al*, 2018). One hundred ng of DNA was digested at 37°C for 12 h with 20 units of *MspI* and *TaqI* (New England Biolabs) in 30 μl of 1× NEB buffer 2. Fill-in and A-tailing were performed by addition of Klenow fragment 3′ > 5′ exo- (New

England Biolabs) and dNTP mix (10 mM dATP, 1 mM dCTP, 1 mM dGTP). After ligation to methylated Illumina TruSeq LT v2 adaptors using Quick Ligase (New England Biolabs), the libraries were size selected by performing a 0.75× cleanup with AMPure XP beads (Beckman Coulter). The libraries were pooled in equal amounts based on qPCR data and bisulfite converted using the EZ DNA Methylation Direct Kit (Zymo Research). Bisulfite-converted libraries were enriched, and quality control was performed using Qubit dsDNA HS (Life Technologies). Fragment length was assessed using high-sensitivity DNA chips on a Bioanalyzer 2000 (Agilent). Sequencing was performed on an Illumina HiSeq 3,000/4,000 instrument in single-end 50 bp mode. Raw *fastq* files were processed within the *NextFlow methylseq* pipeline (Dockerfile v.1.3) with –aligner bismark –singleEnd –genome mm10 –rrbs parameters to obtain methylation coverage. Next, RnBeads (Muller *et al*, 2019) were used for quality control and initial analysis of the DNA methylation data according to established practices (Bock, 2012). We summarized CpGs in 1,000 kb bins and performed differential DNA methylation analysis between amphetamine treatment ($n = 5$) versus control ($n = 6$) or male ($n = 5$) versus female ($n = 6$) datasets using *limma* (Ritchie *et al*, 2015) (FDR-adjusted $P$-value < 0.05, fold change > 1.5, absolute difference > 25 percentage points). The relative over-representation of hypermethylated and hypomethylated tiles was used for GO pathway enrichment analysis (Falcon & Gentleman, 2007). Additionally, we performed base-pair level and region level (for promoters) analysis using *methylKit* (Akalin *et al*, 2012) on Bismark methylation call output files. We used both Fisher's exact test on pooled samples and the $\chi^2$ test with overdispersion correction on covariates (sex or treatment factor) to independently compare sample sets against examined factors (Wreczycka *et al*, 2017). To show a lack of intersections, we visualized six sets of comparisons and CpG island relatedness for all obtained methylation events with *UpSetR* (Conway *et al*, 2017).

### Statistics

All experiments were performed in triplicate unless stated otherwise. Data were expressed as means ± SD or means ± SEM (as indicated). Fold changes represent the percentage change from the mean control value in individual experiments. A $P$-value of < 0.05 was considered statistically significant. Except for analysis of sex differences in 6-week-old animals (histochemistry, GTT, and body weight), all samples were pooled. Differences between treated (amphetamine, methamphetamine, cocaine) and control groups were analyzed using one-way ANOVA (*post hoc* pair-wise comparison). Data from $Ca^{2+}$-imaging experiments were statistically assessed by paired Student's $t$-test. Pearson correlation was used to assess linkage between pairs of parameters.

## Data availability

Interactive genome browser tracks for all genes detected (DNA methylation/RRBS analysis) are available at the UCSC portal (http://bit.ly/AMPHpanDifMet) as well as in GEO (accession no. GSE140072).

**Expanded View** for this article is available online.

## Acknowledgements

The authors thank A. Gavalas and E. Rodriguez (Paul Langerhans Institute Dresden) for training in isolating and culturing embryonic pancreata, Å. Segerstolpe and R. Sandberg (Karolinska Institutet) for access to single-cell RNA-seq data, the Biomedical Sequencing Facility at the Center for Molecular Medicine of the Austrian Academy of Sciences for assistance with next-generation sequencing, and H. Reither, F. Girach, V. Cinquina, and J. Beiersdorf (Medical University of Vienna) for technical assistance. Support for this study was provided by the Swedish Medical Research Council (L.G., T.G.M.H., and T.H.), Novo Nordisk Foundation (T.G.M.H. and T.H.), the European Research Council (ERC advanced grant "Secret-Cells", T.H.), the Swedish Brain Foundation (T.H.), Diabetes Wellness Grant (#720-858-16; R.B.P.), the Austrian Science Fund (FWF; P28683 and P30461, D.D.P. and a Doctoral Program; DOC 33-B27, E.O.T.) and intramural funding from the Medical University of Vienna (T.H.).

## Author contributions

TH conceived the general idea of research; KM, MF, RAR, LG, TGMH, and TH designed experiments; GL, CB, DDP, LG, TGMH, and TH procured funding for experimental work; SK, MK, KM, RBP, EOT, VC, SR, VG, JP, SH, JS, SNR, AZ, OA, IA, EK, and CP performed experiments and analyzed data; and JM provided unique reagents. TH wrote the manuscript with input from all authors.

## Conflict of interest

The authors declare that they have no conflict of interest.

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
