## [Review Process File · The EMBO Journal]

Life-long impairment of glucose homeostasis upon prenatal exposure to psychostimulants

Solomiia Korchynska, Maria Krassnitzer, Katarzyna Malenczyk, Rashmi B. Prasad, Evgenii O. Tretiakov, Sabah Rehman, Valentina Cinquina, Victoria Gernedl, Matthias Farlik, Julian Petersen, Sophia Hannes, Julia Schachenhofer, Sonali N. Reisinger, Alice Zambon, Olof Asplund, Isabella Artner, Erik Keimpema, Gert Lubec, Jan Mulder, Christoph Bock, Daniela D. Pollak, Roman A. Romanov, Christian Piffl, Leif Groop, Tomas G.M. Hökfelt & Tibor Harkany

Review timeline:

Submission date:	19th Oct 2018
Editorial Correspondence:	20th Nov 2018
Editorial Decision:	4th Dec 2018
Revision received:	9th Aug 2019
Editorial Decision:	11th Sep 2019
Revision received:	10th Oct 2019
Accepted:	21st Oct 2019

Editor: Karin Dumstrei

Transaction Report:

Editorial Correspondence

20th Nov 2018

Thanks for submitting your manuscript to The EMBO Journal. Your study has now been seen by three referees and their comments are provided below.

The referees all find the study interesting and in principle suitable for publication here. However, lots of further experiments would be needed in order to consider here. One key point is to add further data to support that psychostimulants act via serotonin signaling. Referee #3 also would like to see more mechanistic insight to explain the long-term effects of the drugs like looking at epigenetic modifications and a more in depth analysis of the mice. I don't know if you have looked at epigenetic marks on some of the potential target areas?? To undertake a genome wide analysis of epigenetic marks I am not so sure would be needed.

All the experiments are in principle doable, but as I don't know how much data you have on hand or can generate within a reasonable time frame of 3-6 months I find it most productive to ask for a point-by-point response before taking a decision on the paper.

REFeree REPORTS:

Referee #1

This study considers the important, lifelong consequences that prenatal exposure to psychostimulants can have on glucose metabolism. There are some interesting findings presented in this manuscript, but there are several issues regarding experimental design, data analysis, and interpretation/discussion of the results that undermine some of the main messages of the study.

Major concerns

No causal role of 5-HT mechanisms is established in the effect of prenatal psychostimulants on β cell function. The authors mention this in their discussion. But even in cases where additional experimental conditions could begin to address this issue, it has not been considered. For example, for Figs. 4B and EV3F, it is necessary to pre-incubate in escitalopram and test whether this occludes the inhibitory effect of the psychostimulants, and vice versa. Prenatal escitalopram exposure should be conducted to establish whether this produces the same changes to β cell function and gene expression as psychostimulant exposure; this is even referred to as a wider implication at the end of the discussion. Without causal experiments, the 5-HT-related observations could be wholly incidental. The authors should take more care when they refer to such things in the text; in section 5 of Results, they say their data is "suggesting a role for FEV in determining physiological insulin availability in humans, too," but there is no evidence that would lead to this claim.

All of the experiments implicating 5-HT relevance consider only protein and gene expression. Since an important aim of this paper is to characterise a 5-HT-centric mechanism of prenatal psychostimulant influence on glucose metabolism, functional analyses of the 5-HT system are needed to support these points.

Proper consideration for the range of psychostimulant targets is not given in this study. For example, the authors choose to exclude the possibility that 5-HT receptor mechanisms are involved by showing an effect brought about by escitalopram (Fig. EV3F); however, this could certainly involve downstream 5-HT receptor involvement. This assumption is applied to multiple experiments throughout the study, and only serves to obfuscate the underlying mechanism. In section 5 of Results, the authors state that their findings suggest "a different mechanism of action for for psychostimulants (e.g. SERT engagement) and 5-HT (receptor engagement)," but this seems contradictory. These 5-HT mechanisms should be clarified with more specificity.

It is difficult to identify the patterns in the representative images of Fig. 2 that are summarised in the graphs and outlined in the text. It would be helpful if better images could be provided. This is particularly important since the semi-quantification of the signal itself seems to have been based on the visual judgement of the author.

On a related note, it is not clear why the authors opted for a semi-quantification of the fluorescent signals. It would be more meaningful if data were reported as numbers/percentages of immunopositive cells. If the authors would like to make comparisons of the signal intensities, this should be done through appropriate image analysis software.

Given that the authors have attempted to quantify and compare the intensities of fluorescent signals across experimental conditions (Fig. 2), the section thickness across experimental conditions needs to be kept consistent. Instead, sections were either 14 μ m (adult and P0) or 8 μ m (E14.5) thick. I don't think it is appropriate to quantify and compare the signals in this situation.

Minor concerns

It is difficult to follow the order in which the results are presented. The authors report mRNA expression from human samples, then continue with prenatal psychostimulant administration in mice, then return to human mRNA, and then back to functional consequences of prenatal psychostimulant administration.

The authors state that "the pathophysiological sign of [psychostimulant] use is uniform, and reminisces that of escitalopram." However, this is based only on calcium changes, and is therefore an overstatement. It should be adjusted accordingly.

If males and females show different changes in Insulin and FEV as a consequence of psychostimulant exposure, how do both sexes exhibit reduced glucose tolerance (Fig. 5)? This suggests that FEV/Insulin are unlikely to be the underlying mechanisms.

There are a few results where the effects of cocaine seem to be different from the other psychostimulants (Fig. 3F3, 4A1, EV6F); do the authors have any thoughts about this?

It is not clear how blockade of SERT by psychostimulants would lead to SERT downregulation (Fig. EV2C) and increased 5-HT in β cells (Fig. 3G-G3).

The Tph2 band in Fig. EV2A isn't really visible.

In section 2 of the results, the authors state, "5-HTR1A immunoreactivity was high at E14.5 and showed inverse correlation with advancing age," but such a correlation is never reported statistically.

Between Figs. 2 and EV2, some comparisons of protein and mRNA expression are made. This could be more helpful if the comparisons included all relevant genes/proteins for section 2 in Results.

It is not clear which experimental group is being analysed in Fig. 4E.

For Fig. 4H1, it is perhaps too strong to say "FEV and GCG expression are unrelated" when the p-value is 0.064. The correlation may be underpowered.

The error bars and statistics for Fig. 5E are not clear.

N numbers in Fig. EV3A are hugely variable.

Referee #2

In this study, Malenczyk et. al, describe a role for in utero exposure to psychostimulants on glucose metabolism in adult off-spring in mice. The authors suggest that psychostimulants act via regulating serotonin signaling in islets. The idea that prenatal exposure to psychostimulants elicits long-lasting metabolic dysfunction in off-spring is very intriguing. The data that psychostimulant delivery (methamphetamine, amphetamine, cocaine) to mothers results in reduced insulin transcript levels and impaired glucose tolerance, specifically in female offspring, is interesting and convincing. However, there are major concerns with the methodology, the rigor of the analyses, and strong conclusions that are drawn based on flawed methodology. The data that psychostimulants act primarily via serotonin signaling is also not very convincing.

Major concerns:

1. The rationale for zooming in serotonin (5-HT) signaling as being the basis for the action of psychostimulants was not clear. The authors ruled out dopamine signaling (which is known to be elevated by psychostimulants) based on profiling for dopamine receptors or transporters from fetal human pancreata. It was not clear why norepinephrine (NE), also a target of psychostimulants, was ruled out. Pancreatic islets are known to secrete and respond to catecholamines. The conclusions that psychostimulants affect islet functions via serotonin but not dopamine or NE should be supported by more rigorous analyses using available mouse models to manipulate these pathways or well-established pharmacological tools.
2. A major concern throughout the manuscript was the use of immunohistochemistry to quantify changes in insulin, glucagon, or 5-HT. The authors should use standard measures in the field such as ELISAs. Similarly, correlation analyses were used to link changes in insulin and Fev (a transcription factor that regulates 5-HT expression)-it is not clear how these correlation analyses were conducted and what they mean functionally.
3. Based on calcium imaging done in a cell line, the authors state that psychostimulants acutely affect glucose-induced insulin secretion in islet beta-cells. It is difficult to extrapolate this acute effect performed in a cell line to understand how in utero drug treatment of mothers elicits long-term effects on glucose homeostasis in adult offspring.
4. The in vitro GSIS experiments suggest an islet intrinsic effect of psychostimulant drugs. However, the results, buried in the supplement, have significant methodological flaws. The islets

were pooled regardless of sex, yet the authors show sex differences in insulin immunoreactivity in stimulant-treated animals, suggesting a need to control for sex. The authors also show a more severe glucose tolerance defect in females than in males, and indeed cocaine seems to have no effect on males at all. Therefore, it is not clear why male and female islets were pooled to look at insulin and glucagon secretion. Also, how do the authors explain that amphetamine exposure had no effect on insulin secretion but had the most robust effect on glucose tolerance.

Additionally, there are basic controls necessary for the *in vitro* GSIS assays. There were no comparisons done between low vs high glucose in control islets. Just eye-balling, the glucose responsiveness of the islets seems weak (Fig. EV6E). Additionally, the data is normalized to islet number which isn't optimal. It would be better to normalize to islet insulin content, and then report the data as a ratio of secreted insulin to insulin content.

5. One of the core points of the study is that prenatal psychostimulant exposure drives dysregulation of serotonin levels, leading to a decrease in FEV levels. However, psychostimulants are known to increase serotonin levels by blockade of re-uptake, and there are reports that elevation of serotonin increases FEV levels (e.g. Kim et al. 2014, the authors cite Henderson and McMillen 1993 which describes the opposite). By what mechanism do the authors propose that prenatal exposure to cocaine directly affects FEV levels?

6. The paper was densely written, that it was hard to follow. The authors used sentences such as "we used bulk mRNA from pancreatic islets of humans stratified for glucose intolerance"-what does stratified for glucose intolerance mean?

Referee #3

This study of Malenczyk et al. is of significant general interest because it provides some evidence to explain the detrimental effects of early-life exposure to psychostimulants on adult metabolism. This has a substantial medical and social impact. Previous publications showed the risk of diabetes in the exposed fetus, although no mechanisms have been provided. It was speculated that the effects could be the result of malnutrition of the fetus, secondary to the effects of the drugs on appetite and nutrition of the mother. This is not the case in the present study, where the body-weight of the offspring of drug-exposed mice is normal. Instead, they describe a new mechanism pointing to direct effects of the drugs on the serotonin signaling and insulin expression in beta cells. The serotonin effects are mediated by the transcriptional regulation of insulin in beta cells. This is a conceptually original and well-conducted study, but more molecular mechanisms are, however, required to support the conclusions of the authors. In particular, I have the following suggestions and concerns:

1. There are two possibilities to explain the long-term effects of the drugs on the future development of type II diabetes of the exposed fetus. First, toxicity effect, which the authors proved that is not the case since mice have normal beta cells mass. The second possibility is cited in the discussion, which is epigenetic modifications. The analyses of such epigenetic changes would provide with more precise molecular mechanisms and would improve the quality of the manuscript.

2. The authors show the analysis of the single cell RNA seq. in the human fetal pancreas. A more in-depth analysis of changes in gene expression in the mouse models used in the study would provide more relevant information related to the phenotype of the mice.

3. All over the manuscript, the authors quantify insulin levels in response to the psychostimulants in cell lines or cellular explants, such as in figure 4. The results would be more physiologically relevant if insulin secretion were stimulated with glucose in the absence or the presence of the drugs. Moreover, insulin protein quantification would be more accurate.

4. This reviewer is confused about the results presented in figure 5 showing that Fev and insulin intensity are decreased by the use of the drugs tested, which correlates with impaired glucose tolerance, indicating a negative correlation between Fev expression and insulin resistance in these mice. However, in figure EV4 the authors show a positive correlation, in humans, of Fev expression and type II diabetes.

5. In addition to the experiments using isolated primary cells, the authors show impaired glucose tolerance in mice exposed to the drugs during fetal development and no differences in body weight. A more in-depth metabolic phenotype of these mice would provide support to the in vitro and ex-vivo findings and give more physiological relevance to the study. Food intake analyses, body fat and lean content, fasting glycemia, fasting insulinemia, insulin sensitivity are some factors that could be analyzed. Moreover, the differences that are shown in the IPGTT are minor (and the number of mice treated is not enough). Feeding mice with a high-fat diet would challenge the pancreas to produce more insulin. In this scenario, the mice exposed to the psychostimulants would have much less insulin secretion.

1st Editorial Decision

4th Dec 2018

Thanks for sending me the point-by-point response. I have now had a chance to take a careful look at everything.

I appreciate the suggested experiments and find that they will strengthen the analysis. I would therefore like to invite you to submit a suitably revised manuscript performing the experiments as outlined in your response. I can also extend the revision time to 6 months.

I take it that for most of your comments that you don't need any further input from me - except for the following

Ref #1 Q1: prenatal escitalopram treatment: I honestly don't know what to advise you on this one. Having analysis in SERT null mice combined with the INS1E experiments +/- escitalopram will clearly strengthen the findings and I tend to agree with you that this bypasses the need to do the prenatal escitalopram treatment. If you are planning to do the NET inhibitor experiments then it would sort of make sense to do the prenatal escitalopram treatment as well. If these experiments are very time consuming and costly to do then I think it is OK not to do them. If on the other hand these are fairly straightforward experiments then they would certainly nicely round off the analysis.

Ref #3 Q2: I agree no need to do single-cell RNA seq in the mouse models. Maybe as you suggested do qPCR for some of the highest expressed genes, but not all.

Ref #3 Q5: I agree with your response.

I think the above points were where you needed input from me. Let me know if I missed something and if you need feedback from me on the other points.

1st Revision - authors' response

9th Aug 2019

Please see next page.

RESPONSES TO REFEREE #1:

We thank the Referee for her/his positive and supportive attitude, as well as constructive queries regarding our original submission. We agree with many of the points raised, and therefore have addressed these experimentally, when possible. Please find our answers as follows.

Q1: 'No causal role of 5-HT mechanisms is established in the effect of prenatal psychostimulants on β cell function. The authors mention this in their discussion. But even in cases where additional experimental conditions could begin to address this issue, it has not been considered. For example, for Figs. 4B and EV3F, it is necessary to pre-incubate in escitalopram and test whether this occludes the inhibitory effect of the psychostimulants, and vice versa. Prenatal escitalopram exposure should be conducted to establish whether this produces the same changes to β cell function and gene expression as psychostimulant exposure; this is even referred to as a wider implication at the end of the discussion. Without causal experiments, the 5-HT-related observations could be wholly incidental. The authors should take more care when they refer to such things in the text; in section 5 of Results, they say their data is "suggesting a role for FEV in determining physiological insulin availability in humans, too," but there is no evidence that would lead to this claim.'

We agree that strengthening causality is important. Moreover, we do agree with the concern that 5-HT effects *per se* were not dealt with in sufficient depth earlier. Therefore, in Figure 2K,K1, we have tested 5-HT effects on INS-1E cells and show that 5-HT reduces the frequency of Ca^{2+} oscillations while increasing their amplitudes (which we interpret as summation of Ca^{2+} transients and generation of large synchronous waves).

Psychostimulant action was tested by determining dose-response relationships on large concentration scales, and in triplicate experiments (Figure 3; see also Figures EV2 & EV3). Firstly, we show that psychostimulants but not escitalopram generate inward currents in cells transfected with SERT (Figure 3A,A1), and that these electrogenic responses are escitalopram sensitive. Secondly, in INS-1E cells, psychostimulants reduce – by and large – both the amplitude and frequency of Ca^{2+} oscillations with cocaine being the exception inasmuch as it increases Ca^{2+} transients at high (100 μ M) concentration. Thirdly, we have tested the approach asked specifically in which the preincubation of escitalopram is followed by superfusion of the psychostimulants (Figure EV2). Considering that escitalopram itself is dose-dependently active on INS-1E cells, we have taken a minimal dose (1 μ M) in which escitalopram is ineffective (Figure 3B1) and combined this with psychostimulants. Our data show that while escitalopram occludes amphetamine effects, it produces summation with cocaine and methamphetamine. This we find logical because we assume that escitalopram's orthosteric binding site only overlaps but is not necessarily identical with that of the drugs and therefore could induce a positive interaction on electrogenic currents.

We have also repeated *in vivo* treatments with escitalopram (in addition to amphetamine) and find that it also disrupts pancreas development in female but not male offspring (Figure 5A,A1). This is not entirely unexpected if we consider that, even if through different mechanisms, both amphetamine and escitalopram block SERT (one by reversal the other by occlusion) and therefore be disruptive. In our discussion, we show parallels with e.g. WIN55,212-2/HU210 (CB1 cannabinoid receptor agonists) and AM251/THC (antagonists) treatments, which are alike: both produce impairments the former through desensitization the latter through receptor occupancy and occlusion.

Lastly, we used a SERT null model (we crossed SERT-Cre knock-in mice to phenotypically SERT null offspring; Appendix Figure S3) and compared these to amphetamine-treated wild-type littermates. These data (Figure EV4C) show, in an independent experiment, a close phenocopy in SERT null mice. Please note that in this independent animal cohort α cells were also affected, and are therefore shown.

In sum, we are confident to have the role of 5-HT addressed sufficiently. The manuscript was edited for more careful wording. Please note that the role of PET1/FEV in humans was elaborated on given our human data that are shown in Appendix Figures S5 & S6.

Q2: 'All of the experiments implicating 5-HT relevance consider only protein and gene expression. Since an important aim of this paper is to characterise a 5-HT-centric mechanism of prenatal psychostimulant influence on glucose metabolism, functional analyses of the 5-HT system are needed to support these points.'

Proper consideration for the range of psychostimulant targets is not given in this study. For example, the authors choose to exclude the possibility that 5-HT receptor mechanisms are involved by showing an effect brought about by escitalopram (Fig. EV3F); however, this could certainly involve downstream 5-HT receptor involvement. This assumption is applied to multiple experiments throughout the study, and only serves to obfuscate the underlying mechanism. In section 5 of Results, the authors state that their findings suggest "a different mechanism of action for psychostimulants (e.g. SERT engagement) and 5-HT (receptor engagement)," but this seems contradictory. These 5-HT mechanisms should be clarified with more specificity.'

We certainly agree with these points. We would however emphasize that receptor engagement is entirely secondary since our *in vitro* assays were done in the absence of extracellular 5-HT. Unless the cells have produced vast amounts of 5-HT relatively instantaneously, it is unlikely that 5-HT receptors were significantly involved acutely. And, even if so, 5-HT receptors will be secondary to amplify/diversify SERT-mediated responses if SERT reversal associates with significant 5-HT efflux. These considerations were dealt with in the revised 'Discussion'.

Yet we have performed all experiments with appropriate dose-response settings (at least three drug doses *in vitro*), and with/without escitalopram (Figure 3 & EV2). Moreover, we used radiolabelled ligand uptake experiments (Figure 2L) to show that rodent-derived INS-1E cells only transport [³H]5-HT but not norepinephrine or dopamine while their reserpin-sensitive VMAT2 activity is negligible. The latter finding confirms data on rodent β -cells in the literature and highlights differences between mouse and human (which we have discussed).

An appealing alternative is to show that differential intracellular 5-HT accumulation upon psychostimulant treatment is central to a mechanism of action. In our 5-HT-loading experiments we found long-lived intracellular 5-HT signal. This is unlikely to be free 5-HT. Instead, we believe this is a reflection of receptor-independent protein serotonylation (which concurs with data from *Tph2*^{-/-} mice published earlier and serotonylation being considered as a key histone modification; both discussed in the revised paper). In Figure 4C, we show data from triplicate experiments in which 5-HT exposure induces intracellular 5-HT signal on proteins while this is abolished when coincidentally administering amphetamine. These data are compatible with amphetamine rendering SERT ineffective to transport 5-HT (whether reversing the transporter or not).

In sum, we view the above as a thorough approach to address the role of 5-HT.

Q3: 'It is difficult to identify the patterns in the representative images of Fig. 2 that are summarised in the graphs and outlined in the text. It would be helpful if better images could be provided. This is particularly important since the semi-quantification of the signal itself seems to have been based on the visual judgement of the author.'

On a related note, it is not clear why the authors opted for a semi-quantification of the fluorescent signals. It would be more meaningful if data were reported as numbers/percentages of immunopositive cells. If the authors would like to make comparisons of the signal intensities, this should be done through appropriate image analysis software.'

Thank you for cautioning us on the image quality and analysis approach. We have significantly changed the data: firstly, we have taken new images for all markers at higher resolution and quality. These data unequivocally show the presence/absence of the immunosignals. We have gone at length analysing co-localization patterns and even deployed 'superresolution' microscopy (Airyscan in Figure 2A1) to justify our conclusions (that is, signals co-exist in cells but they do not arbitrarily co-localize to the same compartments). Moreover, we have removed data on E14.5, which we found most contentious considering the size and early developmental stage of the pancreas.

In addition, we refrained from quantifications in the revised paper. This is because the focus of the paper shifted from a detailed developmental analysis of the pancreatic 5-HT system to the effect of psychostimulants. As such, our anatomical analysis has one single conclusion: 5-HT signalling dominates in β vs. α cells, and this persists throughout mouse development.

Overall, we have streamlined these data such that our conclusions are more to the point and justified.

Q4: 'Given that the authors have attempted to quantify and compare the intensities of fluorescent signals across experimental conditions (Fig. 2), the section thickness across experimental conditions needs to be kept consistent. Instead, sections were either 14 μ m (adult and P0) or 8 μ m (E14.5) thick. I don't think it is appropriate to quantify and compare the signals in this situation.'

Section thickness is an arbitrary variable since we use optical sectioning by high-resolution laser-scanning microscopy. We use uniform 1 μ m-thick optical slices for *post-hoc* image analysis and orthogonal imaging ("z-stacking") to control for equal antibody penetration (which in our experience is only a significant confound in sections > 60 μ m). Nevertheless, by refraining from quantification and only using the anatomical data to pinpoint β cells as a target of psychostimulants the above concerns are alleviated sufficiently.

Q5: 'It is difficult to follow the order in which the results are presented. The authors report mRNA expression from human samples, then continue with prenatal psychostimulant administration in mice, then return to human mRNA, and then back to functional consequences of prenatal psychostimulant administration.'

We have changed the sequence of data presentation to streamline the paper.

Q6: 'The authors state that "the pathophysiological sign of [psychostimulant] use is uniform, and reminisces that of escitalopram." However, this is based only on calcium changes, and is therefore an overstatement. It should be adjusted accordingly.'

This statement has been toned down. Nevertheless, we believe that together with additional data from cellular and mouse models we are in the position to make this claim.

Q7: 'If males and females show different changes in Insulin and FEV as a consequence of psychostimulant exposure, how do both sexes exhibit reduced glucose tolerance (Fig. 5)? This suggests that FEV/Insulin are unlikely to be the underlying mechanisms.'

We believe the figure was misunderstood inasmuch as males do not have altered glucose tolerance. Nevertheless, and in conjunction with the query on epigenetic analysis from Referee #3 we certainly see that FEV is not a primary actuator of the effects we have uncovered. Instead, it is the gene regulatory network upstream from *Pet1/Fev* and insulin that are affected. These data are shown in Figure 7 and discussed extensively.

Q8: 'There are a few results where the effects of cocaine seem to be different from the other psychostimulants (Fig. 3F3, 4A1, EV6F); do the authors have any thoughts about this?'

Indeed, dose-response experiments show that the psychostimulants have significant differences in their active doses and effects. This is not entirely unexpected considering their sites of action and probable SERT-independent effects. These have been discussed.

Q9: 'It is not clear how blockade of SERT by psychostimulants would lead to SERT downregulation (Fig. EV2C) and increased 5-HT in β cells (Fig. 3G-G3).'

SERT downregulation was only seen in explants at E14.5, which is the earliest time-point we have initially tested. However, these data are irrelevant to the treatments that have started later and analysed more advanced developmental stages. Moreover, SERT down-regulation was only robust for methamphetamine at a concentration that induced significant cell death. Therefore, we argue that changes in SERT are a pathological indication rather than a functional one. These data were therefore removed from the paper. We would also note that increased 5-HT content in the cells might not necessarily reflect a change in SERT numbers rather than functionality.

Q10: 'The Tph2 band in Fig. EV2A isn't really visible.'

This figure has been revised.

Q11: 'In section 2 of the results, the authors state, "5-HTR1A immunoreactivity was high at E14.5 and showed inverse correlation with advancing age," but such a correlation is never reported statistically.'

As stated above, these data were removed.

Q12: 'Between Figs. 2 and EV2, some comparisons of protein and mRNA expression are made. This could be more helpful if the comparisons included all relevant genes/proteins for section 2 in Results.'

This figure had a different intent inasmuch as confirming and validating our histochemical results on TPH2 and SERT, and providing confirmation on HTR2b for which we did not have a reliable antibody. Quite certainly, our original phrasing was an over-emphasis of these data, which we have clarified and toned down in the revised manuscript.

Q13: 'It is not clear which experimental group is being analysed in Fig. 4E.'

This analysis includes all conditions since its meaning was to strengthen a correlation between FEV and insulin mRNA levels irrespective of the case condition. This is explicitly stated in the text.

Q14: 'For Fig. 4H1, it is perhaps too strong to say "FEV and GCG expression are unrelated" when the p-value is 0.064. The correlation may be underpowered.'

This statement was revised to state the p value as is.

Q15: 'The error bars and statistics for Fig. 5E are not clear.'

There were no error bars shown to retain maximal visual clarity. Instead, 'area under the curve' calculations were performed to precisely analyse changes in blood glucose. A similar approach was used for the 1-year age group. We have updated the figure legend to alleviate this concern.

Q16: 'N numbers in Fig. EV3A are hugely variable.'

With all due respect, we disagree with the significance of this point since the Referee's comment relates to individual samples, which are further analysed in groups (averaged and normalized per experiment to produce triplicates). Most notably, multitudes of additional data were produced during the revision (including dose-response relationships), which justify both the overall approach and data robustness.

RESPONSES TO REFEREE #2:

We thank the Referee for her/his supportive comments and constructive queries. Please find our point-by-point responses below.

Q1: 'The rationale for zooming in serotonin (5-HT) signaling as being the basis for the action of psychostimulants was not clear. The authors ruled out dopamine signaling (which is known to be elevated by psychostimulants) based on profiling for dopamine receptors or transporters from fetal human pancreata. It was not clear why norepinephrine (NE), also a target of psychostimulants, was ruled out. Pancreatic islets are known to secrete and respond to catecholamines. The conclusions that psychostimulants affect islet functions via serotonin but not dopamine or NE should be supported by more rigorous analyses using available mouse models to manipulate these pathways or well-established pharmacological tools.'

We concur with the Referee's comments to a degree that rigorous analysis is warranted. We have singled out 5-HT signaling because of our *human* fetal and adult transcriptome data. Clearly, a translational element is what we have followed from the human data, which is 5-HT since neither NET nor DAT are expressed at any level comparable to that seen for 5-HT receptors and SERT.

In the revised manuscript, we strengthen this argument by showing that neither norepinephrine nor dopamine is transported in β -cell-like cells (INS-1E; Figure 2L), ruling out a major contribution of the other catecholamines, at least in mouse. These data together with our anatomical results suggest that 5-HT is a primary factor (in both cell-autonomous and intercellular settings) for developing β -cells.

Q2: 'A major concern throughout the manuscript was the use of immunohistochemistry to quantify changes in insulin, glucagon, or 5-HT. The authors should use standard measures in the field such as ELISAs. Similarly, correlation analyses were used to link changes in insulin and Fev (a transcription factor that regulates 5-HT expression)-it is not clear how these correlation analyses were conducted and what they mean functionally.'

Indeed, ELISAs are often used to profile hormone levels in *adult* animals (as we have done it, too). ELISAs are sensitive to target cut-offs, and work reliably only when sufficiently large quantities of tissues/blood are available. In the case of E13.5 or fetal island explants, it is inconceivable to think that ELISAs will pick up any reliable signal. In fact, we have even had problems with HPLC detection of 5-HT in supernatants collected over extended periods of loading (Figure 4B). Therefore, the only viable alternative is the use of histochemistry (in combination with qPCR or *in situ* hybridization for mRNA detection).

We have carefully edited the manuscript to clarify the role of *Pet1/Fev* and its interaction with insulin. Considering that this part of the manuscript is de-emphasized (and moved to Figure EV5 and Appendix Figures S5 & S6), we are confident that the description is now correct.

Q3: 'Based on calcium imaging done in a cell line, the authors state that psychostimulants acutely affect glucose-induced insulin secretion in islet beta-cells. It is difficult to extrapolate this acute effect

performed in a cell line to understand how in utero drug treatment of mothers elicits long-term effects on glucose homeostasis in adult offspring.'

This query leads to the general problematic of linking cellular models to *in vivo* experiments. Here, we have 1) increased the number and condition of mouse models/groups, 2) performed dose-response experiments, 3) used anatomy, functional assays and metabolomics, and 4) notably epigenome analysis in both INS-1E cells and mouse models to argue for the cohesion of our data.

Q4: 'The in vitro GSIS experiments suggest an islet intrinsic effect of psychostimulant drugs. However, the results, buried in the supplement, have significant methodological flaws. The islets were pooled regardless of sex, yet the authors show sex differences in insulin immunoreactivity in stimulant-treated animals, suggesting a need to control for sex. The authors also show a more severe glucose tolerance defect in females than in males, and indeed cocaine seems to have no effect on males at all. Therefore, it is not clear why male and female islets were pooled to look at insulin and glucagon secretion. Also, how do the authors explain that amphetamine exposure had no effect on insulin secretion but had the most robust effect on glucose tolerance.'

Additionally, there are basic controls necessary for the in vitro GSIS assays. There were no comparisons done between low vs high glucose in control islets. Just eye-balling, the glucose responsiveness of the islets seems weak (Fig. EV6E). Additionally, the data is normalized to islet number which isn't optimal. It would be better to normalize to islet insulin content, and then report the data as a ratio of secreted insulin to insulin content.'

We accept that this experiment was suboptimal. In fact, this was to increase islet yield during isolation. We have reported these data (*shown below*), which are not at all integral to our *in vivo* study to link cellular and systemic changes more closely. However, these data can be removed (*only shown below*) without affecting the balance and point of the paper entirely.

(A) Pancreatic islets isolated from 6-week old mice prenatally exposed to methamphetamine or cocaine present lower basal levels of insulin secretion (low glucose) and fail to secrete insulin in response to high glucose stimuli. (B) Only islets isolated from adult mice prenatally exposed to cocaine secrete significantly more glucagon in response to stimulation with low glucose. Data from triplicate experiments were expressed as means \pm s.d.; $n = 30$ islets/group $**p < 0.01$, $*p < 0.05$, two-way ANOVA.

We duly disagree with the statement on controls because control islets properly responded to high glucose challenge: $p = 0.00125$ between low vs. high glucose stimuli. Moreover, secreted insulin was normalized to islet number (a constant factor; broadly accepted in literature) to highlight functional output independent of the cause (decreased insulin content or impaired exocytosis). Therefore, we view this

experimental approach as valid since islet size is unchanged in our *in vivo* experiments at P0.

Q5: 'One of the core points of the study is that prenatal psychostimulant exposure drives dysregulation of serotonin levels, leading to a decrease in FEV levels. However, psychostimulants are known to increase serotonin levels by blockade of re-uptake, and there are reports that elevation of serotonin increases FEV levels (e.g. Kim et al. 2014, the authors cite Henderson and McMillen 1993 which describes the opposite). By what mechanism do the authors propose that prenatal exposure to cocaine directly affects FEV levels?'

We have de-emphasized the role of FEV in our work, which seems as a non-causal correlate of insulin and perhaps one of the upstream regulators that are affected by impaired 5-HT signaling. As such, we show that it is not FEV itself but its upstream gene regulatory network that is affected by psychostimulants. This provides a mechanism linking the various parts of the paper. Please note that we have focused mostly on long-term amphetamine effects in the revised manuscript for this being the most broadly consumed illicit drug by humans.

Q6: The paper was densely written, that it was hard to follow. The authors used sentences such as "we used bulk mRNA from pancreatic islets of humans stratified for glucose intolerance"-what does stratified for glucose intolerance mean?

We have edited the manuscript to increase its appeal. Stratification was described in the Methods, Figure Legends and also highlighted by color-coding in the figures. As stated 'Samples were stratified based on glucose tolerance estimated from HbA1c, i.e. donors with normal glucose tolerance ($HbA1c < 6\%$, $n = 123$), impaired glucose tolerance (IGT, $6\% \leq HbA1c < 6.5\%$, $n = 47$), and type 2 diabetes ($HbA1c \geq 6.5\%$, $n = 32$).'

RESPONSES TO REFEREE #3:

Thank you for your supportive attitude and many constructive thoughts. We certainly agree with you in that the paper could be taken towards different directions, whether being metabolomics or epigenetics. In view of the other Referees' comments, we have made compromise in highlighting epigenetic end-points as most pertinent and valuable to our work. Herein, please find our responses to your specific queries.

Q1: 'There are two possibilities to explain the long-term effects of the drugs on the future development of type II diabetes of the exposed fetus. First, toxicity effect, which the authors proved that is not the case since mice have normal beta cells mass. The second possibility is cited in the discussion, which is epigenetic modifications. The analyses of such epigenetic changes would provide with more precise molecular mechanisms and would improve the quality of the manuscript.'

Indeed, epigenetics is a key opportunity. We have used both INS-1E cells and P0 offspring and processed these for genome-wide DNA methylation profiling and ATAC-seq. We have built an interactive browser to allow anyone to reprocess/probe our dataset (see the revised Methods description). Our data clearly show many hyper- and hypomethylated sites, particularly in those gene regulatory networks that affect serotonin signaling and being upstream from insulin and *Pet1*. Even though our paper is far from being an "epigenetic study" we are confident that these end-points highlight many of the factors that determine, most likely in conjunction, the long-lasting effect we see in cellular and animal models.

Q2: The authors show the analysis of the single cell RNA seq. in the human fetal pancreas. A more in-depth analysis of changes in gene expression in the mouse models used in the study would provide more relevant information related to the phenotype of the mice.

Indeed, this is an exciting opportunity. However, we think that performing single-cell RNA-seq is well beyond the scope of this study. That, in itself, could be a follow-up if combined mouse and human fetal data. Moreover, we are concerned that any single-cell RNA-seq analysis would encounter the complexity of differentiation stages, transdifferentiation, dedifferentiation, ground-state definitions, which will undeniably detract from the message of our manuscript. Instead, we have focused on rationalizing the pharmacology of psychostimulants, and the use of genetic models and extended survival times to clearly indicate gender- and age-specific events involved in psychostimulant action.

Q3: 'All over the manuscript, the authors quantify insulin levels in response to the psychostimulants in cell lines or cellular explants, such as in figure 4. The results would be more physiologically relevant if insulin secretion were stimulated with glucose in the absence or the presence of the drugs. Moreover, insulin protein quantification would be more accurate.'

The data the Reviewer refers to are from *fetal* islets in which a mere few hundred β cells show insulin immunoreactivity. Therefore, ELISAs are insufficient to produce reliable read-outs. As pointed out by Referee #1, we have done the most reliable and sensitive quantification available to use to perform high-resolution analysis of hormone expression in primordial islands in fetal and neonatal mice.

Please also note that the aim of this study is to investigate long-term effects of prenatal exposure to psychostimulants. Therefore, stimulation with drugs during glucose challenge would address a vastly different question.

Q4: 'This reviewer is confused about the results presented in figure 5 showing that Fev and insulin intensity are decreased by the use of the drugs tested, which correlates with impaired glucose tolerance, indicating a negative correlation between Fev expression and insulin resistance in these mice. However, in figure EV4 the authors show a positive correlation, in humans, of Fev expression and type II diabetes.'

Indeed, there is a difference. However, the mice included in our analysis are glucose intolerant but clearly not yet diabetic. In EV4A1 (revised Appendix Figures S5 & S6), the case cohort 'IGT', which could be taken as equivalent to our mouse model does not differ statistically from the control group. It is the T2D subgroup that skews the relationship. Therefore, and even if there are differences between mice and humans, our models do not contradict the human data.

Q5: 'In addition to the experiments using isolated primary cells, the authors show impaired glucose tolerance in mice exposed to the drugs during fetal development and no differences in body weight. A more in-depth metabolic phenotype of these mice would provide support to the in vitro and ex-vivo findings and give more physiological relevance to the study. Food intake analyses, body fat and lean content, fasting glycemia, fasting insulinemia, insulin sensitivity are some factors that could be analyzed. Moreover, the differences that are shown in the IPGTT are minor (and the number of mice treated is not enough). Feeding mice with a high-fat diet would challenge the pancreas to produce more insulin. In this scenario, the mice exposed to the psychostimulants would have much less insulin secretion.'

We agree with the Referee that the data we report are mind-provoking and could lead to many novel lines of research. We also think that these questions are beyond the scope of this study. Likewise, and even though we are experienced with high-fat diet administration, a single experiment would take 3-4 months to complete with additional time required for detailed analysis, and would lead to results that will likely be far-sitting from the scope of this primary manuscript.

To satisfy your question about body weight, we have measured this in a 1-year mouse cohort that we have analysed recently for glucose intolerance (Data in revised Figure 6H), and show that those animals that have impaired glucose tolerance (females prenatally treated with amphetamine) indeed have reduced body weight. These data, together with our epigenome analysis clearly argue for long-term and gender-specific psychostimulant effects.

Thank you for submitting your revised manuscript to the EMBO Journal. Your study has now been re-reviewed by the three referees and their comments are provided below. The referees appreciate the introduced revisions. Referee #3 would like to see a more direct readout of the metabolic phenotype in mice.

The remaining concerns are reasonable, but I also find the study quite complete as is. If there are some straightforward experiments that one can do to address this issue then it would be good to sort this out. If more difficult then lets discuss further.

REFeree REPORTS:

Referee #1:

I am satisfied with the authors' revisions. I have no further comments.

Referee #2:

In this revised manuscript, Korchynska et. al. have addressed several concerns that I raised in the initial submission. Specifically, the authors have provided a stronger rationale for focusing on dysregulated serotonin signaling as a basis for the actions of psychostimulants, instead of catecholamines. However, a caveat is that uptake assays for serotonin, dopamine, and NE were done in a beta-cell like cell line, and not in primary cells. They have also clarified sex differences in the effects of psycho-stimulants on glucose homeostasis. I have a couple of remaining issues that can be clarified by experiments or Discussion.

1. In the initial submission, a common concern of all 3 reviewers was the use of fluorescence intensities from immunostaining analyses to quantify changes in insulin expression. The authors argued that ELISAs are not sensitive enough to probe insulin content in neonatal islets. However, the authors can perform qPCR analyses at these ages. Technically, insulin ELISAs from whole pancreatic extracts and normalized to total protein content should be feasible. At the very least, ELISAs can be used for 6-wk old animals to probe islet insulin content (Fig. 5). These studies would strengthen the mechanism by which maternal exposure to psychostimulants results in islet dysfunction and glucose intolerance in the off-spring.

2. Could the authors clarify why the psychostimulants (for example, amphetamine) treatment of mothers results in glucose intolerance in off spring at 6-weeks (Fig 6C-F) but then results in improved glucose tolerance at 1 yr (Fig, 6I-K)?

Minor issues:

1. The authors seem to be conflicted about the role of Pet1/FEV in the mechanism-they cite previous literature suggesting that serotonin regulates insulin expression via Pet1/FEV as a rationale for the current study. In the revised manuscript, the authors de-emphasize the role of Pet1 in regulating insulin expression upon psychostimulant treatment, but the reasoning is not clear.

2. In some figures, changes in insulin expression is quantified by insulin immunostaining intensity (Fig 4F) while in others, it is represented as insulin-positive cells per section/per islet (Fig 5A1). For the latter, I would consider that as counting the # of beta-cells as opposed to a measure of insulin levels.

Referee #3:

The manuscript from Korchynska et al. has been revised and have incorporated some changes that

were asked by reviewers. As far as the critics and concerns that this reviewer raised, there are still major issues that have not been properly addressed.

First, in question 3 (Q3), it was proposed that insulin secretion experiments were performed in response to glucose challenge. Under physiological conditions, insulin is secreted in response to glucose. This is the only way to measure the function of beta cells in the pancreas. In their study, the authors are measuring insulin content, which is a biologically different process. In the experiments shown in figure 3 the authors use the cell line INS, in which glucose-stimulated insulin secretion experiments could be done. Moreover, single islet insulin secretion experiments and insulin content measurement in fetal pancreata have been previously performed (<https://www.ncbi.nlm.nih.gov/pmc/articles/PMC4708880/>) (<https://www.ncbi.nlm.nih.gov/pmc/articles/PMC5216695/>) using ultra-sensitive ELISA kits.

Second, in question 4 (Q4), the authors say that the mouse data on the correlation between Fev and insulin intensity should be compared to the IGT group of human subjects. Despite there are still big differences in both data sets (a significant correlation in mouse, and not correlation, not even a tendency in humans), this reviewer is still confused about the criteria that the authors used to validate the human data with mouse results.

In question 5 (Q5), there is, first of all, a problem with the experimental design. In most of the in vivo experiments, the number of mice used in the different groups is very heterogeneous. For instance, in figure 6 E, there is one group with 10 mice, the next group 5, the next 8, and the last 3. I do not think that anything can be concluded from these data. In addition, the authors argue that my question is beyond the scope of the study. Well, the aim of the study is to prove that mice exposed prenatally to psychostimulants have impaired lifelong glucose homeostasis. To demonstrate this hypothesis the authors have to show what is the metabolic phenotype of the mice. Only in this way they can prove the physiological relevance of their findings. Therefore, a more in-depth analysis of the metabolic phenotype is important. What happens to insulin secretion tests in mice? Clamp studies? Impaired glucose tolerance may not be related to insulin secretion, but to insulin sensitivity.

Please see next page.

POINT-BY-POINT RESPONSES TO THE REFEREES:

Referee #2:

Q1. “In the initial submission, a common concern of all 3 reviewers was the use of fluorescence intensities from immunostaining analyses to quantify changes in insulin expression. The authors argued that ELISAs are not sensitive enough to probe insulin content in neonatal islets. However, the authors can perform qPCR analyses at these ages. Technically, insulin ELISAs from whole pancreatic extracts and normalized to total protein content should be feasible. At the very least, ELISAs can be used for 6-wk old animals to probe islet insulin content (Fig. 5). These studies would strengthen the mechanism by which maternal exposure to psychostimulants results in islet dysfunction and glucose intolerance in the offspring.”

We appreciate your insistence to this topic. We are of the view that histochemistry will always be more sensitive than ELISA when using the same antibody (given that with histochemistry at super-resolution we can achieve near-single molecule resolution). Nevertheless, we agree with you that qPCRs (assuming that 1) mRNA and protein levels correlate and 2) no post-translational mechanism is involved), are applicable to fetal tissues. We have performed the requested experiments and included the data in Figure EV4. These confirm that amphetamine also reduces insulin mRNA levels.

Q2: “Could the authors clarify why the psychostimulants (for example, amphetamine) treatment of mothers results in glucose intolerance in off spring at 6-weeks (Fig 6C-F) but then results in improved glucose tolerance at 1 yr (Fig, 6I-K)?”

We believe these data can be interpreted in a number of ways. Firstly, baseline glucose levels were different. Secondly, and specifically in females, we did not see a second-phase glucose response. We certainly cannot say that there was improved glucose tolerance at this stage. Instead, there was a difference that might point to a broader metabolic phenotype. Further studies shall reveal if this qualifies as improvement. This is why we concluded that there is a permanent deregulation without quantifying if this observation is benign or adverse.

Q3: “The authors seem to be conflicted about the role of *Pet1*/FEV in the mechanism-they cite previous literature suggesting that serotonin regulates insulin expression via *Pet1*/FEV as a rationale for the current study. In the revised manuscript, the authors de-emphasize the role of *Pet1* in regulating insulin expression upon psychostimulant treatment, but the reasoning is not clear.”

Pet1 is the foremost known transcription factor regulated by insulin signaling in beta cells (it is also expressed in serotonergic neurons actually). This is why we have taken *Pet1* as a prototypic candidate. We do show that *Pet1* levels correlate with that of insulin and are significantly affected by psychostimulants. However, our DNA methylation and ATAC-seq studies do not pin down a direct regulatory change on *Pet1* itself. Instead, they show that upstream gene regulatory networks that impinge upon *Pet1* expression (among other targets) changed. This is why we reached from a candidate gene to an unbiased approach actually, and contrast what is known on *Pet1* with our broader epigenome profiling data.

Q4: “In some figures, changes in insulin expression is quantified by insulin immunostaining intensity (Fig 4F) while in others, it is represented as insulin-positive cells per section/per islet (Fig 5A1). For the latter, I would consider that as counting the # of beta-cells as opposed to a measure of insulin levels.

Please note that we have quantified insulin intensity on the premise that cell numbers did not change (Fig. 4D1,D2). This refined approach showed that despite retained cell numbers there still was a deregulation of insulin production. When the number of beta cells changed, we focused on quantitatively expressing those changes rather than reporting on the secondary analysis of the intensity if immunoreactivity in residual cell cohorts (which might be misleading considering that cell loss and immunoreactivity do not necessarily correlate).

Referee #3:

Q1: “First, in question 3 (Q3), it was proposed that insulin secretion experiments were performed in response to glucose challenge. Under physiological conditions, insulin is secreted in response to glucose. This is the only way to measure the function of beta cells in the pancreas. In their study, the authors are measuring insulin content, which is a biologically different process. In the experiments shown in figure 3 the authors use the cell line INS, in which glucose-stimulated insulin secretion experiments could be done. Moreover, single islet insulin secretion experiments and insulin content measurement in fetal pancreata have been previously performed (<https://www.ncbi.nlm.nih.gov/pmc/articles/PMC4708880/>) <https://www.ncbi.nlm.nih.gov/pmc/articles/PMC5216695/>) using ultra-sensitive ELISA kits.

With all due respect, we wish to remind the Reviewer that insulin release experiments were performed and in following his/her suggestions in the first round of review removed from the paper. However, we refer to our rebuttal letter submitted with the first revision, which contains these data. Therefore, we stand strongly by that besides insulin content also release is affected. Secondly, and as expressed at Reviewer #2/Q1, we are of the view that a combination of mRNA expression and protein analysis is sufficient to support our inferences. We also view histochemistry, if performed precisely, as a “ultra-sensitive” ELISA, the definition of which is rather arbitrary vs. single-molecule resolution by super-resolution microscopy.

Q2: “Second, in question 4 (Q4), the authors say that the mouse data on the correlation between Fev and insulin intensity should be compared to the IGT group of human subjects. Despite there are still big differences in both data sets (a significant correlation in mouse, and not correlation, not even a tendency in humans), this reviewer is still confused about the criteria that the authors used to validate the human data with mouse results.

We do not at any point suggest a 1:1 relevance, equivalence or comparative significance of the two datasets. Yet we find the human data mind-provoking and such that it is supported by our mouse experiments. We deem the two datasets together to credibly emphasize a change in *Pet1* in conditions associated with beta cell dysfunction.

Q3: “In question 5 (Q5), there is, first of all, a problem with the experimental design. In most of the in vivo experiments, the number of mice used in the different groups is very heterogeneous. For instance, in figure 6 E, there is one group with 10 mice, the next group 5, the next 8, and the last 3. I do not think that anything can be concluded from these data. In

addition, the authors argue that my question is beyond the scope of the study. Well, the aim of the study is to prove that mice exposed prenatally to psychostimulants have impaired lifelong glucose homeostasis. To demonstrate this hypothesis the authors have to show what is the metabolic phenotype of the mice. Only in this way they can prove the physiological relevance of their findings. Therefore, a more in-depth analysis of the metabolic phenotype is important. What happens to insulin secretion tests in mice? Clamp studies? Impaired glucose tolerance may not be related to insulin secretion, but to insulin sensitivity.”

Thank you for these points. The experiments the Reviewer refers to are from prenatal treatments and the analysis of offspring from 1) independent pregnancies and 2) in a sex-specific manner. Given biological variations in litter size, indeed the *n*-s are different. These variations are openly and correctly shown in our figures and described throughout. We also emphasize that the background of mice is identical and there was an emphasis on producing biological replicates from independent pregnancies to the highest possible extent. Therefore, we stand firm in our view that the data are valid, both in isolation and in the context of the entire work. Clearly, an ideal pharmacological experimental design would have used identical *n*-s yet statistical analyses account for the number of observations (and degrees of freedom) appropriately.

We have produced a developmentally-oriented and molecular study. We are glad to see that our results are sufficiently interesting to merit potential follow-ups, and we do hope that other laboratories will venture into performing highly sophisticated metabolomic experiments. As a starter to the Referee's line of thought we have tested, using myelin heavy chain (MHC) as marker, if muscle development could be affected in offspring prenatally-exposed to amphetamine. As our data show (see *boxed figure*), MHC expression is significantly reduced ($n = 3$; all P0 females, right forelimb, all muscles; $p < 0.001$) at least by Western blotting. We also have analysed haematoxylin-eosin-stained liver sections and find altered hepatocyte morphology and liver structure. However, these data we deem too preliminary to present. Nevertheless, we view these findings as minimally suggesting that, indeed, prenatal psychostimulant exposure could induce a broad metabolic change, including many organ systems beyond the pancreas, in affected offspring.

Thanks for submitting your revised manuscript to The EMBO Journal. I have now had a chance to take a look at it and all looks good. I am therefore very happy to accept the manuscript for publication here.

Corresponding Author Name: Dr. Tibor Harkany

Journal Submitted to: The EMBO journal

Manuscript Number: EMBOJ-2018-100882